# Experimental Study of Mechanical Properties of Epoxy Compounds Modified with Calcium Carbonate and Carbon after Hygrothermal Exposure

**DOI:** 10.3390/ma13235439

**Published:** 2020-11-29

**Authors:** Anna Rudawska

**Affiliations:** Faculty of Mechanical Engineering, Lublin University of Technology, Nadbystrzycka 36 Str., 20-618 Lublin, Poland; a.rudawska@pollub.pl; Tel.: +48-(81)-5384232

**Keywords:** epoxy compounds, modification, calcium carbonate, carbon, mechanical properties

## Abstract

The objective of this paper is to analyze the effects of hygrothermal exposure on the mechanical properties of epoxy compounds modified with calcium carbonate or carbon fillers. In addition, comparative tests were carried out with the same parameters as hygrothermal exposure, but the epoxy compounds were additionally exposed to thermal shocks. The analysis used cylindrical specimens produced from two different epoxy compounds. The specimens were fabricated from compounds of epoxy resins, based on Bisphenol A (one mixture modified, one unmodified) and a polyamide curing agent. Some of the epoxy compounds were modified with calcium carbonate (CaCO_3_). The remainder were modified with activated carbon (C). Each modifying agent, or filler, was added at a rate of 1 g, 2 g, or 3 g per 100 g of epoxy resin. The effect of the hygrothermal exposure (82 °C temperature and 95% RH humidity) was examined. The effects of thermal shocks, achieved by cycling between 82 °C and −40 °C, on selected mechanical properties of the filler-modified epoxy compounds were investigated. Strength tests were carried out on the cured epoxy compound specimens to determine the shear strength, compression modulus, and compressive strain. The analysis of the results led to the conclusion that the type of tested epoxy compounds and the quantity and type of filler determine the effects of climate chamber aging and thermal shock chamber processing on the compressive strength for the tested epoxy compounds. The different filler quantities, 1–3 g of calcium carbonate (CaCO_3_) or activated carbon (C), determined the strength parameters, with results varying from the reference compounds and the compounds exposure in the climate chamber and thermal shock chamber. The epoxy compounds which contained unmodified epoxy resin achieved a higher strength performance than the epoxy compounds made with modified epoxy resin. In most instances, the epoxy compounds modified with CaCO_3_ had a higher compressive strength than the epoxy compounds modified with C (activated carbon).

## 1. Introduction

Epoxy materials (compounds, adhesives, composites, coatings, cast profiles) have found widespread use in many industries [1,2,3,4,5]. Most applications involve compounds based on epoxy resins and curing agents [1,5,6,7,8]. The curing agents used for epoxy resins are chemical compounds. The compounds trigger three-dimensional cross-linking of the resin, providing it with properties similar to those of epoxy compounds or chemically cured plastics. The selected combination of epoxy resin and curing agent creates a compound with specific properties [7,9]. The properties of epoxy compounds can be modified by adding various modifiers [4,10,11,12,13,14,15,16]. Fillers comprise a considerable group of epoxy compound modifiers. A suitable combination of filler and epoxy resin (with a curing agent) can improve the performance of the cured plastic and reduce, for example, flammability [17,18,19] or its costs [11]. Shaw [20] presented that improvements in high temperature capability of epoxy polymers or epoxy adhesive can be obtained by increasing cross-link density via a suitable choice of resin and/or curing agent. It also facilitates processing by extending the pot life of the epoxy compound while reducing the exothermic effect present during the cross-linking reaction [7]. This requires a suitable combination of the type of filler, filler grain size, and mixing ratio. The type of epoxy and curing agent and the epoxy compound processing technology determine the performance properties of the resulting plastic [3,7].

It is difficult to choose the best of the many commercially available fillers. There is no universal filler that can optimize all performance properties of plastic materials. The selection of a suitable filler should be conditioned by the requirements for the cured resin and its processing. For any specific filler, it should be considered that some performance properties of the plastic material will decrease as other performance properties increase. The addition of a filler to a resin usually improves its mechanical properties. An epoxy resin has a higher tensile strength with the addition of a filler than without it. The bending strength is negatively affected by granular fillers. The compressive strength depends on the filler type [7,11].

One of the popular fillers of epoxy-based polymers is the calcium carbonate (CaCO_3_) filler [11,12,13,21,22,23,24,25]. The form and size of calcium carbonate as a filler vary. Many works concern nanocomposites that contain calcium nano-carbonate as an additive [11,12,23,26]. Some authors [13,25] use calcium carbonate as a filler in epoxy matrix in powder form (e.g., Nwoye et al. [13]; e.g., particle size of between 0.5–0.06 mm, He at al. [12]; e.g., 40–60 nm, Li et al. [11]; e.g., 40 nm with a cubic shape). Powdered calcium carbonate exhibits good dispersion in a different polymer matrix, including an epoxy-base matrix [11]. He and Li [26] presented nanosized calcium carbonate as a reinforcing agent to have good mechanical performance of epoxy matrices. However, an important issue discussed in many studies is obtaining good interfacial adhesion between the matrix, e.g., epoxy and inorganic filler, which results in obtaining good mechanical properties. Surface modification of inorganic materials is essential and necessary to achieve good interfacial adhesion. This aspect was discussed in numerous papers, specifically in the works of Jin and Park [27,28,29] and He at al. [30,31], among others. In addition to surface modification of inorganic fillers, the use of an appropriate filler dispersion method in the matrix is also an important issue. Jojibabu et al. [32] reported that the techniques for dispersion of nanofillers are also important to achieve good mechanical properties of epoxy-based nanocomposites and they presented a comparison of various techniques for dispersion in polymer composites.

Calcium carbonate also improves thermal properties, especially thermal stability [25,27,33]. Al-Zubaidi et al. [25] presented that addition of 15% by weight of CaCO_3_ to the epoxy resin improves the mechanical properties and the thermal resistance. Yang et al. [33] stated that the mechanical properties of the epoxy composites with calcium carbonate were enhanced by the addition of this type of filler. Moreover, the mechanical properties of the modified epoxy composites were enhanced by increasing the amount of calcium carbonate in the epoxy matrix, but they decreased when the filler content reached 2%. The addition of calcium carbonate increases some of the mechanical parameters (such as impact strength and hardness [24,25,33]), but some researchers also observed that some of mechanical properties such as tensile and flexural strength decreased [24]. One of the issues of epoxy resin modification with CaCO_3_ is the amount of filler in the epoxy matrix [21,22]. Scientists use different amounts of this filler in epoxy resin and show different effects of this filler content on the properties of the modified epoxy resin (epoxy composite) [13]. The form of this type of filler is also important. Addition of 15% by weight of calcium carbonate to the epoxy resin was tested by Al-Zubaidi et al. [25]. He et al. [23] showed that a small amount of calcium carbonate, 2–6 wt. %, in the epoxy matrix could significantly increase the thermal stability and mechanical properties. Li et al. [11] presented that the 6% addition of nano-calcium carbonate improves impact strength and flexural modulus. Yang et al. [33] used a form of CaCO_3_ (rod-like and cube-like) with 2% difference in their investigations. Zotti et al. [34] demonstrated that thermomechanical and fracture properties were increased with 10 wt. % of carbon carbonate filler. Maisel and Wason [35] studied the effect of different percentage compositions of silica and CaCO_3_ fillers on the tensile strength of epoxy resin in the range up to 3%.

Another filler of epoxy-based polymers is the carbon (C) filler, especially as carbon fiber reinforced resin (CFRP) [31,32,36,37,38,39]. Carbon as an additive to polymers (fillers) can be in various forms, such as, for example, carbon nanofillers and carbon nanotubes. With regard to the sizes, a nanoscale filler is often used. Gude et al. [38] noticed that addition of carbon nanofillers to the epoxy adhesive had an insignificant effect on the aging behavior. In this work, two types of carbon nanofillers were added to two-component epoxy adhesive: carbon nanofillers (CNFs) and carbon nanotubes (CCNts). Both types of nanofillers were supplied in dry powder form. Although these fillers hinder the adsorption of water in the adhesive, the failure mode in most cases is adhesive or cohesive in the laminates. Consequently, the effect of addition of carbon fillers to the epoxy matrix could not be transparently assessed.

Based on many studies conducted, it is noticed that different results are obtained depending on the amount of the filler, its form, and operating conditions. As epoxy materials are used in various forms (adhesive, composite, product in the form of some structural element, coating, etc.) and are subjected to various factors, it seems that research on modified materials and their properties when exposed to various environmental factors is still needed. Many researchers [40,41,42,43,44,45,46] have conducted experiments on epoxy adhesive and adhesive joints under hygrothermal exposure. Okba et al. [47] rightly noticed that it is necessary to conduct research related to the exposure time and thermal conditions of use of adhesives and adhesive joints of various construction materials, including composites and modified adhesives, as it affects the safety of use of bonded structures. Zhang et al. [48] underlined that the strength of adhesive joints could change after a long-term environmental exposure and they evaluated the rate effect on the strength of adhesively bonded joints (prepared by epoxy adhesive) after hygrothermal exposure (80 °C and 95% RH). Therefore, it seems advisable to undertake research in this direction. Sales et al. [36] investigated the hygrothermal effects on fracture toughness of composite carbon/epoxy joints (80 °C and 90% RH). They underlined that wet and hot environments affect both the adherend and the adhesive. Gude et al. [38] studied the effect of hygrothermal aging at 55 °C and 95% RH on the shear strength of carbon fiber/epoxy laminates adhesive joints. The effects of operating factors such as seasoning in water solution containing iron sulfate and the time of seasoning were also investigated by Rudawska and Brunella [49]. Based on the obtained results, it was found that excessive iron sulfate content in water has a negative effect on the selected mechanical properties of epoxy adhesive compound. In other works by Rudawska et al. [50], Rudawska [51], and Miturska et al. [52], seasoning time was also one of the factors analyzed during aging tests. It was noticed that both temperature and time of seasoning have an impact on the adhesive mechanical properties. It was also observed that the compressive strength of the epoxy compound samples seasoned at ambient temperature increases with the seasoning time increase.

Two epoxy compounds were analyzed, based on Bisphenol A (one epoxy resin modified, and one unmodified) with a polyamide curing agent. One group of the epoxy compound was modified with calcium carbonate (CaCO_3_). The other group was modified with activated carbon (C). Each modifying agent, or filler, was added at a rate of 1 g, 2 g, or 3 g per 100 g of epoxy resin. The selection of the filling quantity was made on the basis of the work presented by He [12] and Yang [33]. The effect of hygrothermal exposure at 82 °C temperature and 95% RH humidity was examined. The effects of thermal shocks, achieved by cycling between 82 °C and −40°C, on selected mechanical properties of the filler-modified epoxy compounds were also investigated.

## 2. Materials and Methods

### 2.1. Unmodified and Modified Epoxy Compounds

For these tests, 12 modified epoxy compounds were selected to create different versions. The versions varied by epoxy resin type, filler type, and filler quantity. The epoxy compounds compounded with Bisphenol A were as follows: unmodified epoxy resin, epoxy number between 0.49 and 0.52 mol/100 g (Epidian 5, Sarzyna Resins, Nowa Sarzyna, Poland), and modified epoxy resin (a mixture of Epidian 5 and a modified polyester resin), epoxy number 0.40 mol/100 g (Epidian 57, Sarzyna Resins, Nowa Sarzyna, Poland), with the unmodified and modified epoxy resins cured with a polyamide curing agent with an amine number between 290 and 360 mg KOH/g (PAC, Sarzyna Resins, Nowa Sarzyna, Poland). The modified epoxy compounds were fabricated with either of two filler types: activated carbon (C) (CWZ-22, Stanlab, Lublin, Poland) or calcium carbonate (CaCO_3_) (Limestone Industry Plants, Trzuskawica S.A., Poland), added in various quantities, resulting in three versions for each of the two filler types used. The typical concentration of CaCO_3_ is 98.23% and its molecular weight is 100.09 g/mol. Activated carbon was used in the form of dust, with a molecular weight of 12.01 g/mol. A total of 12 modified epoxy compounds were tested (2 base epoxy compounds × 2 filler types × 3 filler quantities); see Table 1 and Table 2 for the epoxy compound designations used in this paper. The epoxy compounds ratio was set at 100:80 resin:polyamide curing agent (in weight), corresponding to the stoichiometric epoxy/amide molar ratio.

Based on the information presented in the works [11,13,21,22,23,33,35], it was noticed that scientists use a different amount of the calcium carbonate filler. According to the literature data, the amount of filler may be from 2% to 8% parts by weight. He at al. [23] underlined that performance of polymeric materials can be improved by introducing some small amount of filler <5 wt. %. Yang et al. [33] pointed out that the mechanical properties of the epoxy composites were improved by increasing the amount of CaCO_3_ added, but they decreased when the filler content reached 2%. The addition of 1–3 g of fillers per 100 g of epoxy resin was used in the experiments to prepare the epoxy compounds. The same amount of addition of both fillers was assumed for comparison purposes.

### 2.2. Epoxy Compounds Technology

The modified epoxy compounds were preprocessed first by batching a specific amount of the epoxy resin (100 g), followed by addition of the these materials in the order listed, and at three different weight ratios: 1% filler content per 100 g of the epoxy resin (version 1); 2% filler content per 100 g of the epoxy resin (version 2); 3% filler content per 100 g of the epoxy resin (version 3). The components were hand-mixed with the fillers. The mixing stage continued for approximately 30 s to blend the epoxy resin thoroughly with the respective filler. The second preprocessing stage consisted of adding 80 g of the curing agent per 100 g of each epoxy resin. The resulting compound was mixed for 90 s with a disk agitator at 128 m/min to obtain a homogeneous material. The epoxy compound components were batched with an electronic scale (OX-8100, mfg. FAWAG S.A, Poland, measurement accuracy 0.1 g, ISO 9001) in a polymer–plastic container. The epoxy compound preprocessing was completed at 21 °C ± 2 °C and 21% ± 3% RH.

### 2.3. Description of Epoxy Compounds Specimens

The epoxy compound preprocessing was followed (with due consideration of the pot life) by casting them into polyethylene molds, each of which was cylindrically hollow, with the internal dimensions *d* = 15 ± 0.5 mm and *L* = 60 ± 0.5 mm. The inner surfaces of the polypropylene molds required suitable pretreatment before casting by application of an adhesion-reducing agent, POLSIFORM (Silikony Polskie, Nowa Sarzyna, Poland). The adhesion-reducing agent was sprayed on the inner surfaces from a distance of approximately 30 cm. Three minutes after applying the adhesion-reducing agent, the liquid epoxy compounds (the epoxy compounds) were cast into the molds. The casting was performed at 21 ± 2 °C and 21% to 23% RH. Following the curing process (see Section 2.4), the as-cast epoxy specimen dimensions were measured and determined at diameter *d* = 15 ± 0.5 mm and length *L* = 40 ± 1 mm (Figure 1a). Figure 1b gives an overview of the examples of the as-cast epoxy specimens.

The cured specimens were machined to an equal height on an FNB26 milling machine. The machining process used a six-blade milling cutter (Ø35 mm) made from high speed steel.

### 2.4. Curing and Aging Conditions

The tested specimens were cured in a single run at 21 °C ± 2 °C and 21% ± 3% RH. The overall curing time was 7 days for all specimens. Three batches of the epoxy specimens were fabricated in total and for the variants listed in Table 1 and Table 2. Each batch comprised 72 specimens. The first specimen batch was assigned as reference specimens, and not exposed to any additional effects. The reference specimens were prepared to enable a comparative determination of whether the specified aging conditions would affect the strength of the specimens made from each epoxy compound. Four reference specimen batches were made, each comprising 18 pieces (Table 3).

The scanning electron microscope (SEM) images of examples of modified epoxy compounds samples (reference samples) after the curing process were presented in Figure 2. Scanning electron microscopy using a MIRA 3 TESCAN microscope (TESCAN ORSAY HOLDING, a.s., Brno, Czech Republic) was used to obtain the SEM images.

The microscopic images allowed visualization of the obtained structure of the modified epoxy composition after the curing process (the technological parameters were mentioned above). The noticeable heterogeneous distribution of the filler is probably the result of the mixing method used, as well as the lack of preparation of the filler surface and the type of filler. In the case of the addition of carbon as filler, smaller clusters (agglomerates) were observed in the epoxy matrix (resin and curing agent) than in the case of calcium carbonate. Further analysis of the structure of the epoxy composite will be discussed.

The other two specimen batches were aged. The second batch of epoxy specimens was placed in a climate chamber, Espec SH-661 (ESPEC EUROPE GmbH, distributed by Klimatest, Poland) for 10 weeks. The third batch of epoxy specimens was placed in a thermal shock chamber, Espec TSE-11 (distributed by Klimatest, Poland), for 10 weeks. Table 4 and Table 5 show the aging conditions of the analyzed epoxy compounds.

The test conditions in the thermal shock chamber presented in Table 5 were approximately 110 cycles (+82 °C/−40 °C). The time of 15 min included the passage of the epoxy composition samples (placed in the basket) from the cold chamber to the warm chamber (and vice versa) along with the stabilization temperature of the samples. The exposure time was only counted at the trigger temperatures and the achieved set temperature.

### 2.5. Strength Test

Strength tests were performed following the curing of the reference specimens and both the high-temperature aging and thermal-shock aging of the remaining specimens (second and third lots). All fabricated specimens were tested on a Zwick/Roell Z150 tester (ZwickRoell GmbH&Co. KG, Ulm, Germany). The compressive strength was determined with the test method specified in EN ISO 604 standard. The test parameters were: preloading = 100 N, test load = 10 mm/min, maximum deformation 15%. The tests were focused on the analysis and comparison of compression modulus, compression strength, and compressive strain. The specimens were strength-tested in a special fixture with a sleeve shape, in which cylindrical samples of cured epoxy compounds samples were placed, which, at the same time, ensured the perpendicularity of the sample face to the cylindrical surfaces of the holder.

## 3. Test Results

### 3.1. Reference Specimens Strength Test Results for the Modified Epoxy Compounds

#### 3.1.1. Calcium Carbonate

The results for the versions of the reference specimens made from E5/PAC/100:80 and E57/PAC/100:80 and modified with calcium carbonate are shown in Figure 3 (compression modulus), Figure 4 (compression strength), and Figure 5 and Figure 6 (compressive strain).

The compression modulus of the E5/PAC/100:80 epoxy compound was the highest for the specimens with 3 g of calcium carbonate (806 MPa), with the highest result of the two epoxy compounds (Figure 3). It was observed that this specific epoxy compound had a compression modulus which rose with increasing filler content. The lowest compression modulus was found for the lowest calcium carbonate content for the E5/PAC/100:80 epoxy compound, 237 MPa, and at only 30% of the maximum compression modulus, 806 MPa (achieved for E5/PAC/100:80/CaCO_3_/3%).

The highest compression modulus for the E57/PAC/100:80 epoxy compound was achieved for the 2% calcium carbonate version (554 MPa), although the compression modulus achieved for the 1% filler content was only slightly lower (521 MPa for E57/PAC/100:80/CaCO_3_/1%), with a 6% difference between the results. The lowest compression modulus value (142 MPa) was found for the E57/PAC/100:80/CaCO_3_/3% epoxy version, which was 75% lower than for the Epidian 57 epoxy resin compound.

The results presented in Figure 3 show that the E5/PAC/100:80 epoxy compound had a higher compression modulus than E57/PAC/100:80; extremely high differences of the value were found between the version with the 1% calcium carbonate filler content (55%) and the version with the 3% filler content (83%). A different relationship was also found for the analysis of the compression modulus test results; the result was the lowest for E57/PAC/100:80/CaCO_3_/3% and the highest for E5/PAC/100:80/CaCO_3_/3%. It is probably due to the nature of the interaction (properties) of the matrix and the filler. In the first case, the epoxy resin (Epidian 57) had a higher viscosity than the Episian 5 epoxy resin. This probably contributed to better dispersion and fewer calcium carbonate agglomerates in the lower viscosity resin. Therefore, it can be assumed that the type (properties) of the epoxy resin, which is the basis of the matrix (along with the curing agent), plays an important role in the creation of epoxy composites. Moreover, Li et al. [11] underlined that it is difficult to disperse nanosized particles in highly viscous polymer matrix.

The results in Figure 4 show that E5/PAC/100:80 had a higher compressive strength than E57/PAC/100:80. The maximum tested compressive strength (77 MPa) was found for the specimen lot made from E5/PAC/100:80/CaCO_3_/3%. Of all the specimens made from E5/PAC/100:80, the lowest compressive strength (37 MPa) was found for the E5/PAC/100:80/CaCO_3_/1% specimens. The lowest compressive strength (14 MPa) was found for E57/PAC/100:80/CaCO_3_/3% specimens, and the epoxy compound with the 2% filler content had the highest compressive strength.

An analysis of the results in Figure 5 and Figure 6 shows that the highest compressive strain (9.9%) was developed for E57/PAC/100:80/CaCO_3_/3%. E5/PAC/100:80/CaCO_3_/1% achieved a slightly lower compressive strain (8.8%). Note that the remaining results for E5/PAC/100:80 and E57/PAC/100:80 were similar, between 4.8% and 5.1%. The obtained strength results are also confirmed by the results of the stress values presented in Figure 6. Higher stress values were obtained for the compositions containing the unmodified resin (Epidian 5) compared to the compositions containing the modified resin (Epidian 57).

#### 3.1.2. Activated Carbon

The results for the versions of the reference specimens made from E5/PAC/100:80 and E57/PAC/100:80 and modified with activated carbon (C) are shown in Figure 7 (compression modulus), Figure 8 (compressive strength), and Figure 9 and Figure 10 (compressive strain).

The results in Figure 7 show that compression modulus differed widely between the tested epoxy compounds, at nearly 50% between each compared version. The highest value of *E_C_* (697 MPa) was developed for E5/PAC/100:80/C/1%; the lowest was developed for the version with the 2% filler content (435 MPa), or 62% of the highest tested value. The compression modulus for the 3% filler version was 608 MPa, at 13% lower from the highest tested value. The widest spread of compression modulus occurred with the E5/PAC/100:80/C/3% version. For E57/PAC/100:80, the lowest compression modulus (229 MPa) occurred for E57/PAC/100:80/C/2% (2% activated carbon-filler version). E57/PAC/100:80/C with 1% and 3% of the same filler achieved similar results (331 MPa and 314 MPa, respectively). The difference between the minimum and the maximum values for the epoxy compound (E57/PAC/100:80) was 30%.

An analysis of the results (Figure 8) shows that the highest compressive strength (70 MPa) was developed for the specimen lot made from E5/PAC/100:80 with the 1% filler content per 100 g of the epoxy resin. The lowest compressive strength was found for the E5/PAC/100:80/C 2% activated carbon version. All specimen lots made from E57/PAC/100:80/C achieved much lower compression strengths, with the highest difference being 69% between the epoxy versions made from the other epoxy resin type. In all versions of E57/PAC/100:80/C, the compressive strength was similar.

The compressive strain test results presented in Figure 9 and Figure 10 show that the E57/PAC/100:80/C specimens developed higher compressive strain values than the E5/PAC/100:80/C specimens. The highest compressive strain (8.5%) was found for the E57/PAC/100:80/C/2% version. The lowest compressive strain (4.6%) was found for the E5/PAC/100:80/C/1% version. The highest standard deviation of compressive strain was found for the E5/PAC/100:80/C/2% specimens. Comparing the stress–strain curves (Figure 10) for both types of epoxy compositions, it is noted that, in most cases, the analyzed materials are materials of reduced stiffness, exhibiting some of the characteristics of ductile materials.

### 3.2. Strength Test Results for the Climate-Chamber-Aged Modified Epoxy Compounds

#### 3.2.1. Calcium Carbonate

Figure 11 shows the compression modulus values of the epoxy specimens (E5/PAC/100:80/CaCO_3_ and E57/PAC/100:80/CaCO_3_) after aging in the climate chamber.

The compression modulus values (Figure 11) do not show any significant differences. The highest value of *E_C_* (339 MPa) was found for E5/PAC/100:80/CaCO_3_/3%, at the highest filler content. The lowest value of the same parameter (267 MPa) was developed for E5/PAC/100:80/CaCO_3_/1%, at the lowest filler content. The lowest compression modulus was found for E57/PAC/100:80/CaCO_3_/3% (175 MPa), and the highest value of the same parameter was found for E57/PAC/100:80/CaCO_3_/1%, at the lowest filler content. For the Epidian 5-based epoxy compounds, the compression modulus values increased with increasing filler content; the trend was inversed for the Epidian 57-based epoxy compound.

Figure 12 shows the compressive strengths of the epoxy specimens made from E5/PAC/100:80/CaCO_3_ and E57/PAC/100:80/CaCO_3_ after aging in the climate chamber. The compressive strength test results (Figure 12) for E5/PAC/100:80/CaCO_3_ and E57/PAC/100:80/CaCO_3_ varied slightly between the individual versions. Significantly higher values were developed by the specimen lots made from E5/PAC/100:80/CaCO_3_. The highest compressive strength was for E5/PAC/100:80/CaCO_3_/2% (65 MPa). The compression strengths for E57/PAC/100:80/CaCO_3_ was less than half that for E5/PAC/100:80/CaCO_3_, between 25 MPa and 33 MPa.

Figure 13 and Figure 14 show the compressive strain values of the epoxy specimens made from E5/PAC/100:80/CaCO_3_ and E57/PAC/100:80/CaCO_3_ after aging in the climate chamber.

An analysis of the results presented in Figure 13 and Figure 14 show that the highest and the lowest compressive strain (7.8%) was developed for the E57/PAC/100:80/CaCO_3_/3% specimens. The remaining values in Figure 13 vary slightly from each other. Similar results of compressive strain were found for E5/PAC/100:80/CaCO_3_/1% and E57/PAC/100:80/CaCO_3_/2%. They had the lowest values of this parameter and reached 6%. The highest standard deviation of compressive strain was found for the specimen lots made from E5/PAC/100:80/CaCO_3_/3%, at a value of about 31%. After aging in a climate chamber (Figure 14), it can be seen that the stress–strain shape for more ductile materials was observed in the case of modified CaCO_3_ epoxy compounds containing modified epoxy resin (Epidian 57) than in compounds in which Epidian 5 epoxy resin was used as a matrix. These differences were observed in compared to reference samples (Figure 6). Therefore, it may be assumed that the higher temperature showed the differences in the compounds containing different resins.

#### 3.2.2. Activated Carbon

The comparison of the strength-tested parameters compression modulus, compressive strength, and compressive strain for E5/PAC/100:80/C and E57/PAC/100:80/C after aging in the climate chamber is shown, respectively, in Figure 15, Figure 16, Figure 17 and Figure 18.

An analysis of the results in Figure 15 shows that the highest compression modulus (281 MPa) was developed for the E5/PAC/100:80/C/1% specimens; for the 3% filler version, the compression modulus was the lowest (228 MPa), with a difference of 19%. For the Epidian 57-based epoxy compounds, the lowest compression modulus was found for E57/PAC/100:80/C/3%, 155 MPa; the lowest compression modulus value was found for the 1% filler versions, 281 MPa. The spread between the values was 43%. The highest standard deviation of compression modulus was found for the E5/PAC/100:80/C/3% version. In both modified epoxy compounds, the compression modulus decreased while the filler (activated carbon) content increased.

Figure 16 presents a comparison of the compression modulus between the E5/PAC/100:80E57/PAC/100:80 epoxy compounds modified by the activated carbon filler after aging in the climate chamber.

The compression strengths for both epoxy compounds (E5/PAC/100:80/C and E57/PAC/100:80/C) showed wide differences of up to 50%. The E57/PAC/100:80/C specimens achieved poorer results for compression strength. These were between 20 MPa and 33 MPa. As a matter of comparison, the difference between the highest and the lowest compression strengths in Figure 16 was 62%. The highest compressive strength was developed for E5/PAC/100:80/C/3% at 53 MPa. It differed insignificantly from the 1% filler version, which achieved 52 MPa. The lowest spread of compressive strength test results was for the 3% filler versions made from E5/PAC/100:80/C/3% (1.5%).

Figure 17 and Figure 18 show the change trend of compressive strain of the epoxy specimens made from E5/PAC/100:80/C and E57/PAC/100:80/C after aging in the climate chamber.

The results (Figure 17) show a trend inverse to those presented in Figure 15 and Figure 16. The specimen lots made from E57/PAC/100:80/C had higher compressive strain than the E5/PAC/100:80/C specimens. The highest compressive strain value in Figure 17 was 9.9% for the E5/PAC/100:80/C/2% and E57/PAC/100:80/C/3% epoxy compounds. The lowest compressive strain (6.6%) was found for the E57/PAC/100:80/C/1% version. The difference between the maximum and the minimum compressive strain was 33% for the Epidian 57-based epoxy compounds and 5–9% for the Epidian 5-based epoxy compounds. The highest standard deviation of compressive strain was 26% (E5/PAC/100:80/C/2%), and the lowest compressive strain was 1% (E57/PAC/100:80/C/2%). Analyzing the results presented in the Figure 18, similar relationships were noted as for epoxy compounds containing the calcium carbonate filler (Figure 14).

### 3.3. Strength Test Results for the Thermal-Shock-Chamber-Aged Modified Epoxy Compounds

#### 3.3.1. Calcium Carbonate

Figure 19, Figure 20, Figure 21 and Figure 22 show the strength test results for the epoxy specimens, with the CaCO_3_ filler, aged in the thermal shock chamber.

The results presented in Figure 19 show that the maximum and minimum compression modulus values were developed for the epoxy samples made from E5/PAC/100:80/CaCO_3_/2% (420 MPa) and E5/PAC/100:80/CaCO_3_/1% (400 MPa), respectively. The difference was 5%. The results for the E57/PAC/100:80/CaCO_3_ specimens were lower than for E5/PAC/100:80/CaCO_3_. The lowest compression modulus value found was 229 MPa (E57/PAC/100:80/CaCO_3_/2%), or 60% of the maximum for the parameter. The standard deviation of compression modulus for E5/PAC/100:80/CaCO_3_/2% was relatively high (33%) in comparison to the other results.

Figure 20 shows a comparison of the compressive strength test results for E5/PAC/100:80/CaCO_3_ and E57/PAC/100:80/CaCO_3_.

A comparison of the compression strengths of the epoxy specimens made from E5/PAC/100:80/CaCO_3_ and E57/PAC/100:80/CaCO_3_ shows large differences (Figure 20). The compressive strength test results for the E57/PAC/100:80/CaCO_3_ versions were similar at 22 MPa, 19 MPa, and 21 MPa, respectively. No effect of the filler content for the epoxy compound on compressive strength was found. The highest compressive strength was for E5/PAC/100:80/CaCO_3_/2% (79 MPa). The lowest result for the same epoxy compound (49 MPa) was found for E5/PAC/100:80/CaCO_3_/3%, and the difference between the maximum and the minimum value was 38%. The lowest spread (8%) of the compressive strength test results was found for E57/PAC/100:80/CaCO_3_/3%.

Figure 21 and Figure 22 show the compressive strain of the epoxy specimens made from E5/PAC/100:80/CaCO_3_ and E57/PAC/100:80/CaCO_3_.

The results in Figure 21 show that the compressive strain values were not in significant variance to each other; in one instance, they were identical (E5/PAC/100:80/CaCO_3_/1% and E57/PAC/100:80/CaCO_3_/1%). E57/PAC/100:80/CaCO_3_/2% achieved the highest compressive strain (8.5%). The lowest result (5.6%) was found for E5/PAC/100:80/CaCO_3_/2%. The standard deviation of compressive strain was the highest (30%) for E57/PAC/100:80/CaCO_3_/1%. It can be seen (Figure 22) that in the case of the three variants of the epoxy compounds containing calcium carbonate, in which the matrix is the unmodified Epidian 5 resin, the stress–strain curve takes the form of a curve for elastic materials with forced elasticity (materials with reduced stiffness). However, for the three variants of the modified CaCO3 compounds containing the modified resin (Epidian 57), these curves are characteristic for ductile materials. On this basis, it can be indicated how important the selection of the matrix (type of epoxy resin) is in modified epoxy compositions, which allows obtaining different properties of such compositions.

#### 3.3.2. Activated Carbon

Figure 23, Figure 24, Figure 25 and Figure 26 show a comparison of the strength test results for the epoxy specimens made from E5/PAC/100:80/C and E57/PAC/100:80/C, both modified by activated carbon as the filler.

A study of the results in Figure 23 shows that the highest compression modulus was developed for E5/PAC/100:80/C/3%. The value was 595 MPa. The lowest compressive strength was found for E57/PAC/100:80/C/3% (3 g of filler). The value was 142 MPa. The maximum to minimum difference was 76%. The E5/PAC/100:80/C/3% epoxy version showed that as the filler content increased, so did the compression modulus. The maximum to minimum difference was 44%. The highest spread of the test results (21%) was found for the specimen lot made from E5/PAC/100:80/C/1%.

Figure 24 shows a comparison of the compressive strength test results for E5/PAC/100:80/C and E57/PAC/100:80/C.

The results for compressive strength for E5/PAC/100:80/C and E57/PAC/100:80/C varied by more than 50% between the compounds. For E5/PAC/100:80/C, the highest compressive strength was for the E5/PAC/100:80/C/3% version with the 3 g filler content. The value was 59 MPa, and the highest result achieved for the parameter of all the epoxy versions shown in Figure 24. The E5/PAC/100:80/C epoxy versions demonstrated that as the filler content increased, the compression strength also increased. The maximum to minimum difference was 27% for the results. For E57/PAC/100:80/C, the lowest result was found for E57/PAC/100:80/C/3% (14 MPa). The remaining results for the same epoxy compound were only slightly higher: 21 MPa for E57/PAC/100:80/C/1% and 23 MPa for E57/PAC/100:80/C/2%. The highest standard deviation of compressive strength was found for the E5/PAC/100:80 epoxy versions with 1 g of activated carbon (E5/PAC/100:80/C/1%) and 2 g of activated carbon (E7/PAC/100:80/C/2%): 15% and 12%, respectively.

Figure 25 and Figure 26 show a comparison of the compressive strain test results for E5/PAC/100:80/C and E57/PAC/100:80/C.

A comparison of results (Figure 25 and Figure 26) shows that the highest compressive strain, 5.7%, was found for E5/PAC/100:80/C/3%. The lowest result (4.9%) was developed for the E5/PAC/100:80/C/1% specimens. The compressive strain values were similar (and identical in some instances) for the E57/PAC/100:80/C versions. The values were close to one another, at 5% for the 1 g activated carbon version (E57/PAC/100:80/C/1%), 5.2% for the 2 g activated carbon version (E57/PAC/100:80/C/2%), and 5.6% for the 3 g activated carbon version (E57/PAC/100:80/C/3%). The highest standard deviation of compressive strain was found for the specimens made from E5/PAC/100:80/C/35%. It can be noticed (similar to the epoxy compounds modified with calcium carbonate) that the stress–strain curves (Figure 26) in the case of modified epoxy compounds with activated carbon, containing unmodified resin (Epidian 5) as a matrix in all three types of this compound, but with different filler content, takes the form of a curve for materials with reduced stiffness (elastic with forced elasticity). However, for the three variants of the compound modified with carbon, but containing the modified resin (Epidian 57), these curves are characteristic for ductile materials, although with elongations of a dozen or so percent, the ductility is rather moderate.

## 4. Comparative Analysis of Results

The comparative analysis of the results was based on the compressive strength test results related to the function of filler (modifier) content on the tested epoxy compounds. Figure 27 lists the compressive strength test results for the reference specimens, the climate-chamber-aged specimens, and the thermal-shock-aged specimens. The E5/PAC/100:80 and E57/PAC/100:80 epoxy compounds were modified by 1 g of calcium carbonate and activated carbon as the filler, respectively.

The results for compressive strength shown in Figure 27 for the 1% filler epoxy compounds, which were aged under different conditions, showed the relations discussed below.

For E57/PAC/100:80/CaCO_3_/1%, the exposure to both high temperature and thermal shocks (cycling) reduced the compressive strength in comparison to the reference specimens. The compressive strength was reduced by 27% post-high temperature and 35% post-thermal shocks.The high temperature did not reduce the compressive strength for E57/PAC/100:80/C/1%, which was 33% higher than for the reference specimens. There was also no decrease in the compressive strength value of this modified epoxy compound subjected to the thermal shocks.It is believed that this specific epoxy version benefited in terms of resistance to the tested operating exposures from the modification with activated carbon.For E5/PAC/100:80 (based on unmodified Epidian 5), the type of filler (at 1%) produced completely different relations. The CaCO_3_ modified epoxy compound had a higher compressive strength after high temperature exposure than the reference specimens (+37%); the post-thermal shock of the same specimen compound had an even higher compressive strength (+54% vs. the reference specimens). Here, CaCO_3_ was beneficial to the compressive strength of the epoxy compounds exposed to high temperatures and thermal shocks. An inverse relationship was found for the same E5/PAC/100:80 epoxy compound when modified by activated carbon. In this case, the heat temperature and thermal shocks reduced the compression strength.The modified E5/PAC/100:80 epoxy compounds had a higher compressive strength in comparison to E57/PAC/100:80 for every compared version of the filler type and quantity and the effects of the operating exposure.

The list of the compressive strength test results for the E5/PAC/100:80 and E57/PAC/100:80 epoxy specimens modified with the 2 g filler content (both CaCO_3_ and C) are shown in Figure 28.

An analysis of the results listed in Figure 28 shows the following:The comparison of E5/PAC/100:80 to E57/PAC/100:80 show that, yet again, E57/PAC/100:80 had the lower compression strength.The epoxy compounds modified with CaCO_3_ had a higher compressive strength than the epoxy compounds modified with C (activated carbon) (with the exception of one epoxy version).The compressive strength for E57/PAC/100:80 with 2% of CaCO_3_ demonstrated a negative effect of the filler after aging. Compared to the reference specimens, the high-temperature aging reduced the compressive strength by 30%; the thermal shocks reduced the compressive strength by 52%. The modification with activated carbon did not negatively affect the compression strength, but slightly improved it. However, for most Epidian 57-based epoxy resin compounds with activated carbon as the filler, the compressive strength was lower than for the compounds modified by calcium carbonate.A comparison of the E5/PAC/100:80 specimens aged in the climate chamber (high-temperature aging) to the reference samples showed a reduction in compressive strength for both modifiers.For the same type of E5/PAC/100:80, the thermal shocks did not reduce the compressive strength in comparison to the reference specimens, irrespective of the modifier. The compressive strength of the same epoxy compound exposed to thermal shocks was higher than for the epoxy compounds exposed to high-temperature aging.

Figure 29 shows a comparison of the compressive strength test results for E5/PAC/100:80/C and E57/PAC/100:80/C with the 3 g filler content.

An analysis of the results listed in Figure 29 shows the following:(1)For the reference samples:
(a)The compressive strength of the unmodified E5/PAC/100:80 specimens was much higher than for the E57/PAC/100:80 specimens.(b)The highest compressive strength (77 MPa) was found for E5/PAC/100:80/CaCO_3_/3% (modified with calcium carbonate).(c)The lowest compressive strength (14 MPa) was found for E57/PAC/100:80/CaCO_3_.(d)The difference between the highest and the lowest compression strengths was 82% in the reference specimens.(2)For the high-temperature (climate chamber) aged specimens:
(a)The highest compressive strength (61 MPa) was found (yet again) for E5/PAC/100:80/CaCO_3_/3%. A slightly lower value (53 MPa) was found for the E5/PAC/100:80/C/3% specimens.(b)The E57/PAC/100:80 specimens also achieved poorer results. The lowest value (20 MPa) was found for the E57/PAC/100:80/C/3% (modified with activated carbon).(c)The plotted difference between the highest and the lowest compressive strength was 67%.(3)Comparison of the epoxy specimens aged in the thermal shock chamber:
(a)The highest compressive strength (59 MPa) was found for E5/PAC/100:80/C/3% (modified with activated carbon).(b)A lower value (49 MPa) was found for the same epoxy compound (E5/PAC/100:80/CaCO_3_/3%, modified with calcium carbonate).(c)Low compressive strengths were identified for the specimens based on E57/PAC/100:80. The epoxy specimen lot modified with calcium carbonate (E57/PAC/100:80/CaCO_3_/3%) achieved a compressive strength of 21 MPa, while the specimens modified with activated carbon (E5/PAC/100:80/C/3%) achieved 14 MPa.(d)The maximum to minimum difference was 76%.

To conclude on the compressive strength test results for the 3% filler epoxy compounds after aging under different conditions:▪The epoxy compounds modified with calcium carbonate had a higher compressive strength in most instances (10 out of 12 epoxy versions), irrespective of the aging method.▪The epoxy compounds with the basic unmodified Epidian 5 resin had a compressive strength over two times higher than any of the compared specimen versions, irrespective of the modifier type or aging method.▪For E5/PAC/100:80, the high-temperature aging reduced the compressive strength for the 3% CaCO_3_ versions (−20%) and the C versions (approximately −16%).▪The aged E5/PAC/100:80 modified with activated carbon and CaCO_3_ reduced the compressive strength by 6% and 34%, respectively, in comparison to the reference compounds.▪For E57/PAC/100:80/CaCO_3_, the high-temperature and thermal shock aging methods did not affect the compression strength; the high-temperature aging increased the parameter value by a factor of two.▪The epoxy compounds based on Epidian 57 and modified with activated carbon suffered from thermal shock aging, with an approximately 30% reduction in compressive strength compared to the reference specimens.

The visual inspection of the behavior of the specimens during the strength tests demonstrated that unmodified epoxy resin (Epidian 5) contained polyamide curing agent fabricated an epoxy compounds which frequently cracked during leveling of the edges and in the strength tests. The epoxy specimens made from E57/PAC/100:80 were much more elastic. It was necessary to fabricate several lots of the specimens, as their edges suffered from strong deformations when leveled. The aging of the specimens improved their hardness. In general, it can be concluded that in the case of the modified epoxy compounds, the observed failure of the samples, in most cases, showed the brittle failure characteristics. These results were also noticed in other works by the author [51].

Summarizing the results presented on compression stress–strain curves, it can be noted that for some samples, these curves exhibit a large discrepancy. Probably, one of the reasons for such a discrepancy could be some fluctuations in the climatic conditions during the tests or the different curing process (despite maintaining the same technological parameters for the preparation of the compounds) of epoxy compositions containing a different type of epoxy resin (unmodified epoxy resin Epidian 5 and modified epoxy resin Epidian 57). These results require further analysis.

Okba et al. [47] stated that the decrease of residual compressive and tensile strengths depends on, e.g., the type of adhesive, elevated temperature level, and to a lesser effect, on exposure time. The authors also presented that the bond strength decreased as the thermal exposure level and exposure time increased with various contributions of each parameter on the strength. Rudawska [51] underlined that the addition of calcium carbonate to the epoxy compounds (epoxy resin and curing agent) contributes to a higher compressive strength as the aging time increases. A reduction in the brittleness of the resin may be the cause of these results. The results presented also by Banea et al. [53] show that as the temperature increases the adhesive tensile strength reduces but the ductility increases. The results presented by Miturska et al. [52] indicated that the type of filler affects the strength of the tested compositions over the exposure time. In the case of montmorillonite, the strength increases, but as far as addition of carbon and chalk is concerned, the strength decreases over the exposure time for tested epoxy resin compounds. Li et al. [11] indicated that impact strength of the nano-CaCO_3_ toughened epoxy resin composite increases with an increasing nano-calcium carbonate content in the range of less than 6 wt. %, while impact of strength of the composite declines when filler content is more than 6 wt. %. He et al. [12] concluded that small contents (2–6 wt. %) of nano-calcium carbonate in the epoxy compound can increase the mechanical properties and thermal stability of these composites, whereas Yang et al. [33] indicated that the mechanical properties of the CaCO_3_/epoxy composites were enhanced by increasing the amount of calcium carbonate added, but they decreased when the filler content reached 2%. Jin and Park [28] revealed that the fracture toughness and impact strength of trifunctional epoxy resin/calcium carbonate nanocomposites were significantly increased by addition of nano-calcium carbonate filler. Therefore, it can be assumed that both the type of filler, its amount, type of epoxy resin, and aging conditions require further research, especially for the application of such composite materials. The variety of ingredients and aging conditions is still an interesting source of research.

Although the study did not analyze the method of surface preparation of fillers, many scientists emphasize this aspect of modification of polymer composites (including epoxy) as important due to the obtained mechanical properties. He at al. [31] demonstrated that the treated carbon fibers composites could possess excellent interfacial properties with mixed resin and could improve interlaminar shear strength. In another work, He at al. [12] pointed out that the improvement of thermal and mechanical properties is attributed to the surface modification of nano-fillers (e.g., nano-calcium carbonate), which can enhance the interfacial properties between nano-calcium carbonate fillers and epoxy resin.

## 5. Conclusions

Strength testing was performed on specimens made from four different epoxy compounds. The specimens were made from unmodified epoxy resin and modified epoxy resin cured with polyamide curing agent. The resulting epoxy compounds (E5/PAC/100:80 and E57/PAC/100:80) were modified with calcium carbonate and activated carbon as the filler. An important exposure factor in the tests was the seasoning of some of the epoxy specimens in a climate chamber and the remainder in a thermal shock chamber for approximately 10 weeks in both cases.

An analysis of the results led to the conclusion that the type of tested epoxy compound and the quantity and type of filler determine the effect of the climate chamber aging and the thermal shock chamber processing on the compressive strength for the tested epoxy compounds. It was difficult to explicitly determine the effect of high-temperature aging (in the climate chamber) and thermal-shock aging (in the thermal shock chamber) on the strength parameters of the specimens. In some of the examined cases, the effects of both high temperature and thermal shocks negatively affected the compressive strength of the epoxy compounds, while it was increased for some of the modified epoxy compounds. The observations discussed here apply to other mechanical parameters such as compression modulus and compressive strain. The epoxy compounds which contained unmodified epoxy resin (Epidian 5) achieved a higher strength performance than the epoxy compounds made with modified epoxy resin (Epidian 57). It can be assumed that this is related to the properties of the epoxy resin. The lower viscosity resin (Epidian 5) produced better filler dispersibility than the higher viscosity resin (Epidian 57). In most instances, the epoxy compounds modified with CaCO_3_ had a higher compressive strength than the epoxy compounds modified with C (activated carbon). The different quantities of the fillers, 1–3 g of calcium carbonate (CaCO_3_) and activated carbon (C), largely determined the strength parameters, with results differing from the reference specimens and the epoxy specimens aged in the climate chamber and the thermal shock chamber. The influence of aging in a climatic chamber and thermal shocks chamber on the change of the mechanical properties of modified epoxy compounds containing unmodified resin was noticed. After aging in the chambers, these epoxy compounds showed features of a material with reduced stiffness (elastic with forced flexibility), and before aging, they showed features of a more plastic material.

In summary, proper compounding of an epoxy adhesive compound significantly determines its strength. Aging can also affect the mechanical properties of epoxy compounds.

## Figures and Tables

**Figure 1 materials-13-05439-f001:**
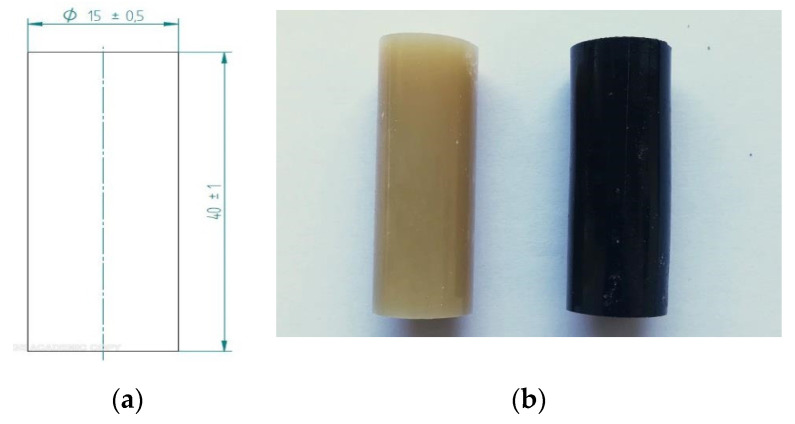
(**a**) Epoxy specimen dimensions (unit: mm); (**b**) cured epoxy specimens: E57/PAC/100:80/CaCO_3_ (**left**), E57/PAC/100:80/C (**right**).

**Figure 2 materials-13-05439-f002:**
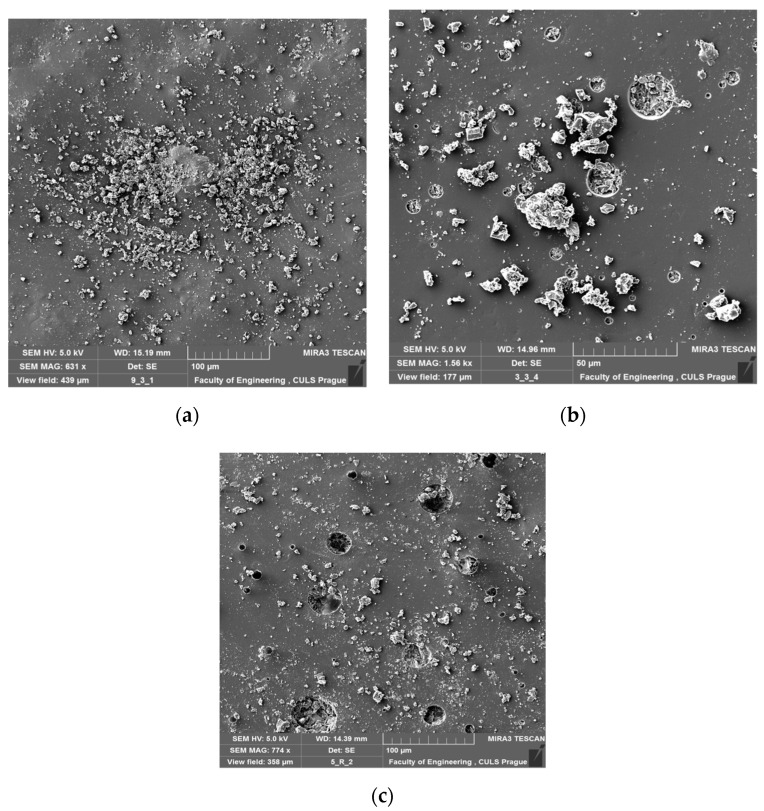
SEM images of modified epoxy compounds after curing process: (**a**) and (**b**) E57/PAC/100:80/ CaCO_3_/2%, (**c**) E57/PAC/C/2%.

**Figure 3 materials-13-05439-f003:**
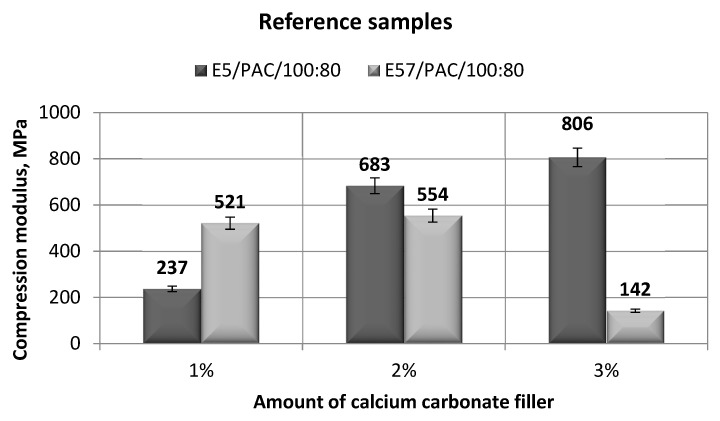
Comparison of the compression modulus between the E5/PAC/100:80 and E57/PAC/100:80 epoxy specimens modified by the calcium carbonate filler.

**Figure 4 materials-13-05439-f004:**
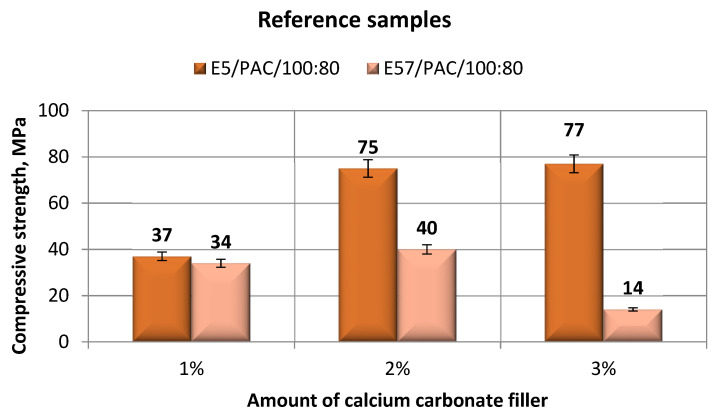
Comparison of the compressive strength between the E5/PAC/100:80 and E57/PAC/100:80 epoxy specimens modified by the calcium carbonate filler.

**Figure 5 materials-13-05439-f005:**
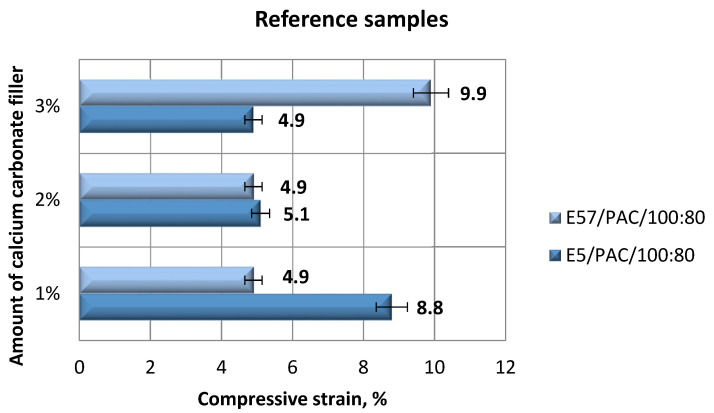
Comparison of the compressive strain between the E5/PAC/100:80 and E57/PAC/100:80 epoxy specimens modified by the calcium carbonate filler.

**Figure 6 materials-13-05439-f006:**
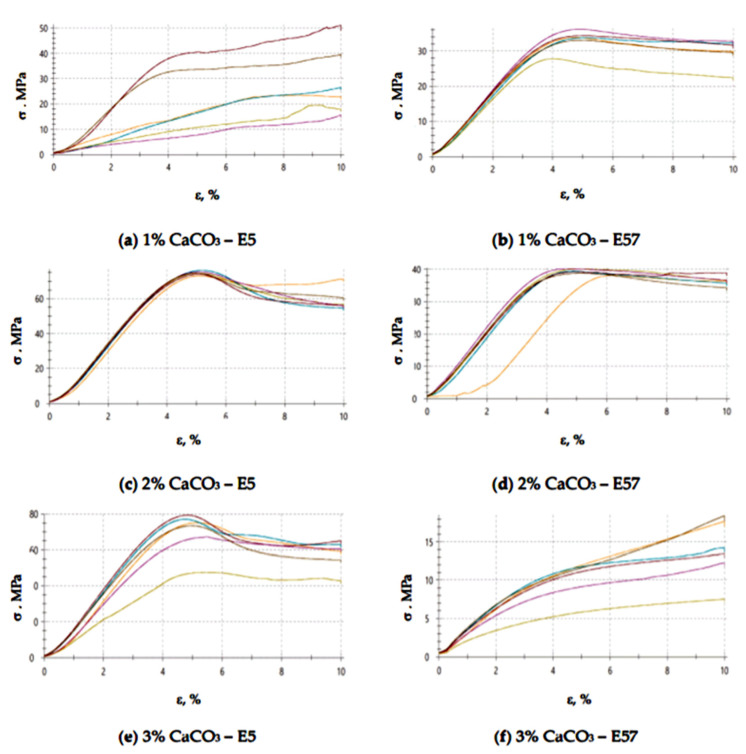
Stress–strain curves of the E5/PAC/100:80 and E57/PAC/100:80 epoxy specimens modified by the calcium carbonate filler: (**a**) 1% CaCO_3_—E5, (**b**) 1% CaCO_3_—E57, (**c**) 2% CaCO_3_—E5, (**d**) 2% CaCO_3_—E57, (**e**) 3% CaCO_3_—E5, (**f**) 3% CaCO_3_—E57.

**Figure 7 materials-13-05439-f007:**
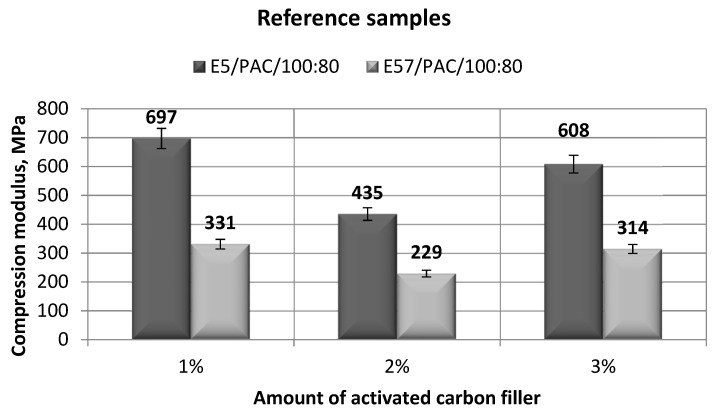
Comparison of the compression modulus between the E5/PAC/100:80 and E57/PAC/100:80 epoxy specimens modified by the activated carbon filler.

**Figure 8 materials-13-05439-f008:**
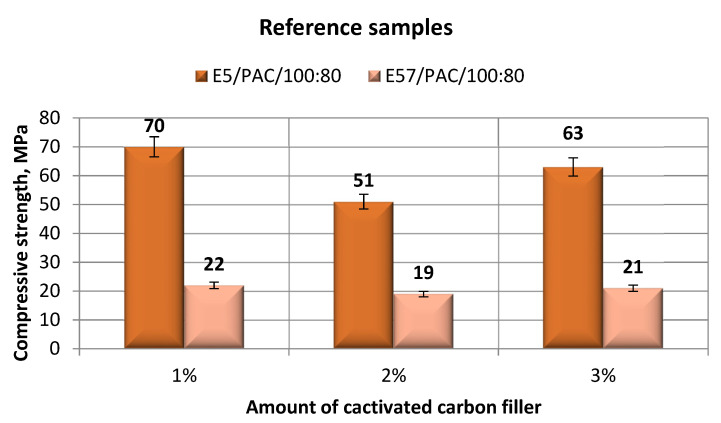
Comparison of the compressive strength between the E5/PAC/100:80 and E57/PAC/100:80 epoxy specimens modified by the activated carbon filler.

**Figure 9 materials-13-05439-f009:**
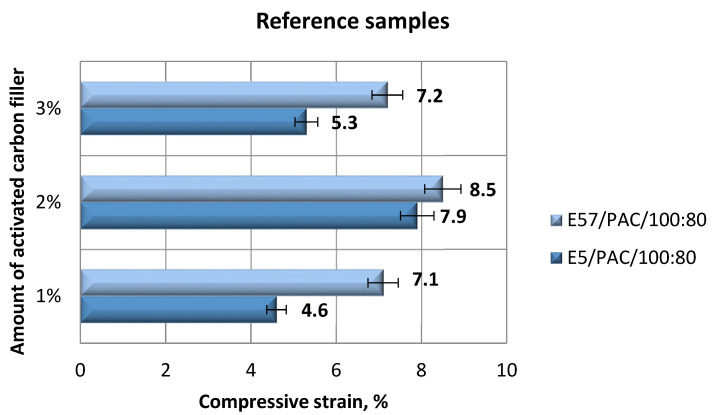
Comparison of the compressive strain between the E5/PAC/100:80 and E57/PAC/100:80 epoxy specimens modified by the activated carbon filler.

**Figure 10 materials-13-05439-f010:**
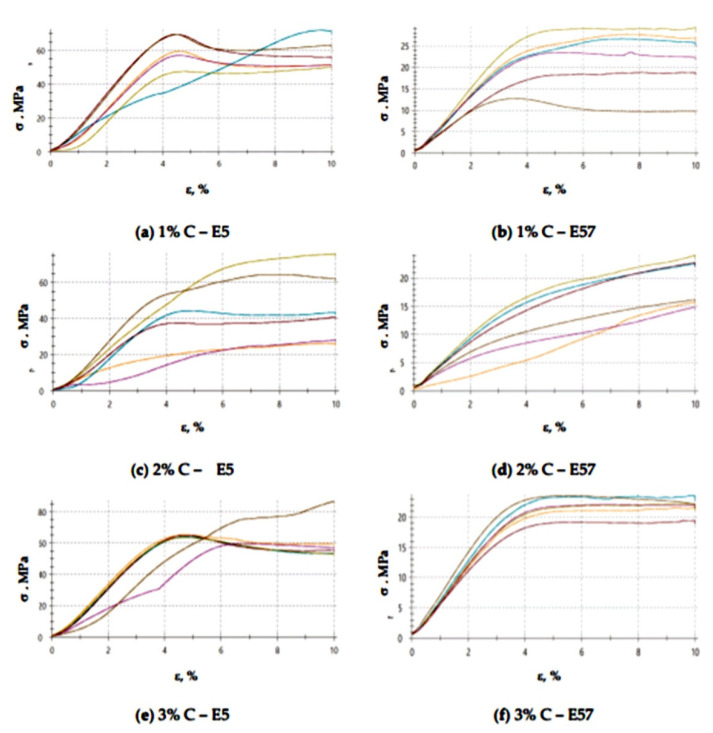
Stress–strain curves of the E5/PAC/100:80 and E57/PAC/100:80 epoxy specimens modified by the activated carbonate filler: (**a**) 1% C—E5, (**b**) 1% C—E57, (**c**) 2% C—E5, (**d**) 2% C—E57, (**e**) 3% C—E5, (**f**) 3% C—E57.

**Figure 11 materials-13-05439-f011:**
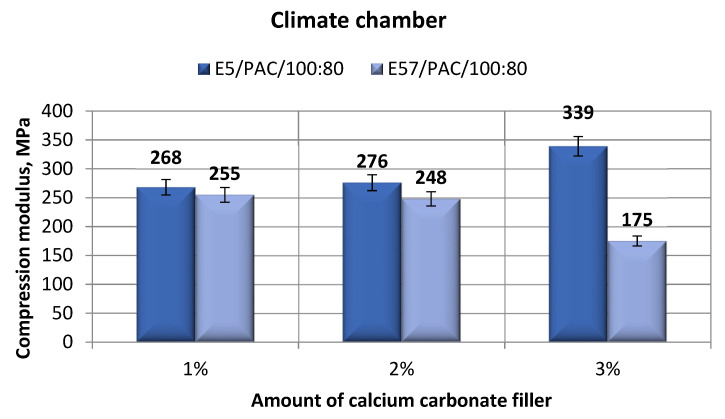
Comparison of the compression modulus between the E5/PAC/100:80 and E57/PAC/100:80 epoxy specimens modified by the calcium carbonate filler post-climate chamber aging.

**Figure 12 materials-13-05439-f012:**
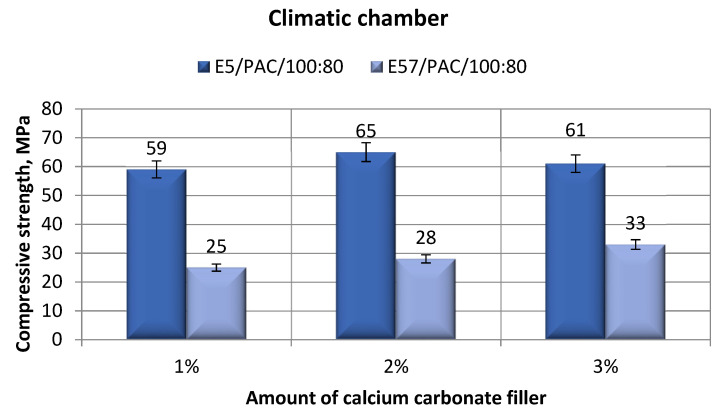
Comparison of the compressive strength between the E5/PAC/100:80 and E57/PAC/100:80 epoxy specimens modified by the calcium carbonate filler post-climate chamber aging.

**Figure 13 materials-13-05439-f013:**
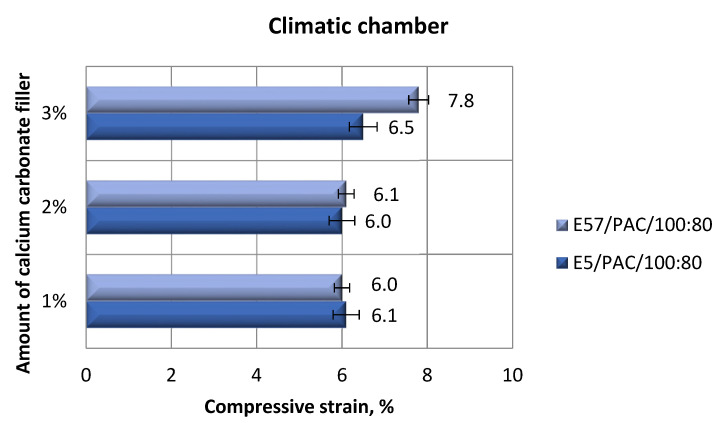
Comparison of the compressive strain between the E5/PAC/100:80 and E57/PAC/100:80 epoxy specimens modified by the calcium carbonate filler post-climate chamber aging.

**Figure 14 materials-13-05439-f014:**
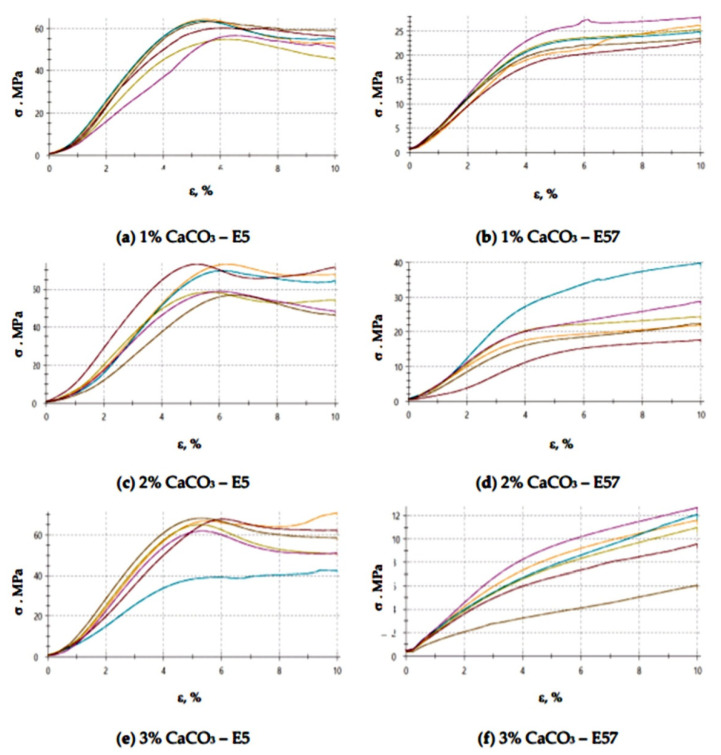
Stress–strain curves of the E5/PAC/100:80 and E57/PAC/100:80 epoxy specimens modified by the calcium carbonate filler post-climate chamber aging: (**a**) 1% CaCO_3_—E5, (**b**) 1% CaCO_3_—E57, (**c**) 2% CaCO_3_—E5, (**d**) 2% CaCO_3_—E57, (**e**) 3% CaCO_3_—E5, (**f**) 3% CaCO_3_—E57.

**Figure 15 materials-13-05439-f015:**
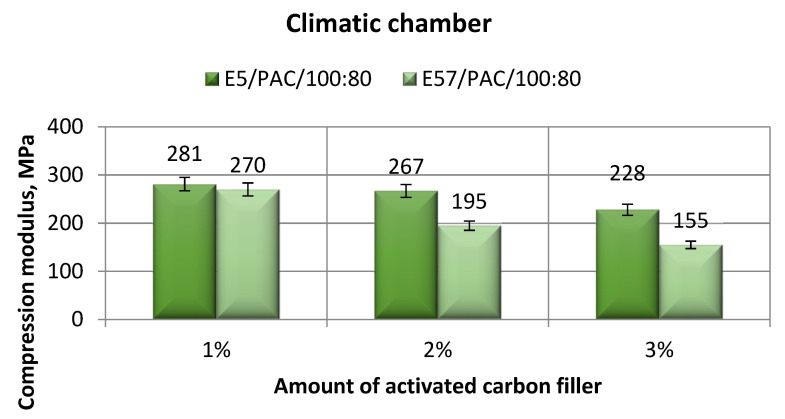
Comparison of the compression modulus between the E5/PAC/100:80 and E57/PAC/100:80 epoxy specimens modified by the activated carbon filler post-climate chamber aging.

**Figure 16 materials-13-05439-f016:**
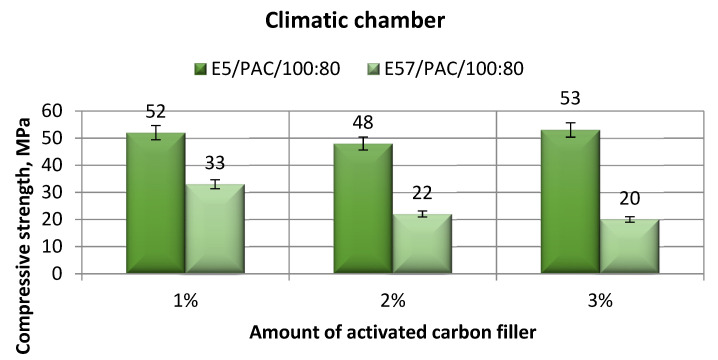
Comparison of the compressive strength between the E5/PAC/100:80 and E57/PAC/100:80 epoxy specimens modified by the activated carbon filler post-climate chamber aging.

**Figure 17 materials-13-05439-f017:**
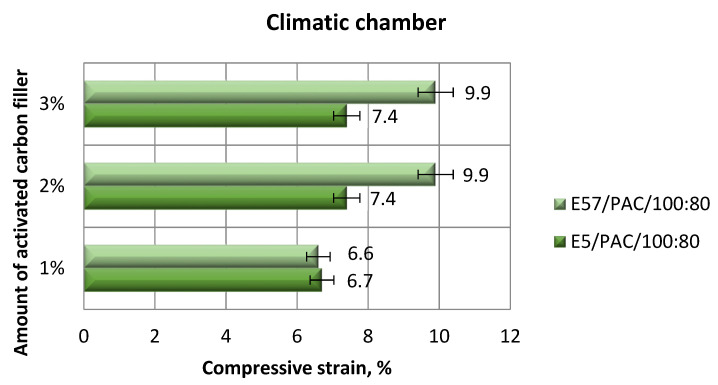
Comparison of the compressive strain between the E5/PAC/100:80 and E57/PAC/100:80 epoxy specimens modified by the activated carbon filler post-climate chamber aging.

**Figure 18 materials-13-05439-f018:**
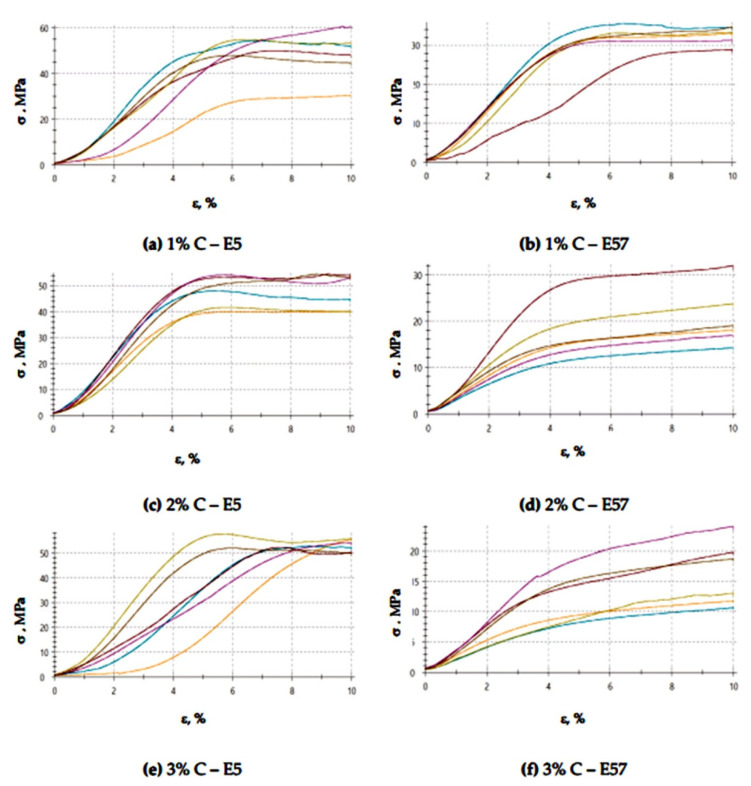
Stress–strain curves of the E5/PAC/100:80 and E57/PAC/100:80 epoxy specimens modified by the activated carbonate filler post-climate chamber aging: (**a**) 1% C—E5, (**b**) 1% C—E57, (**c**) 2% C—E5, (**d**) 2% C—E57, (**e**) 3% C—E5, (**f**) 3% C—E57.

**Figure 19 materials-13-05439-f019:**
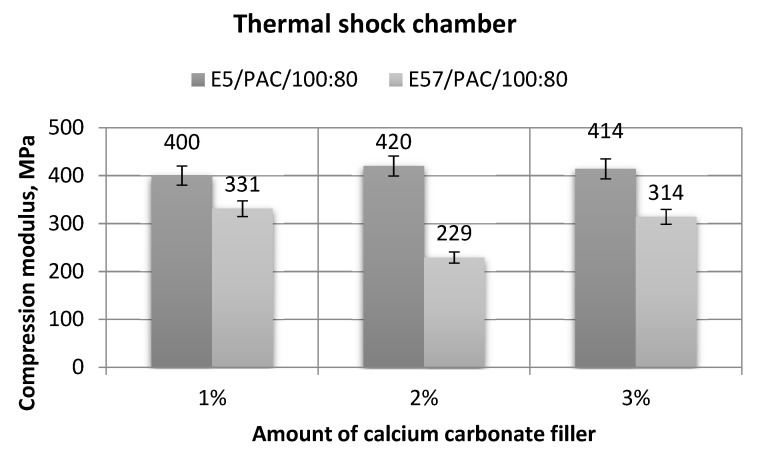
Comparison of the compression modulus between the E5/PAC/100:80 and E57/PAC/100:80 epoxy specimens modified by the calcium carbonate filler post-thermal shock chamber aging.

**Figure 20 materials-13-05439-f020:**
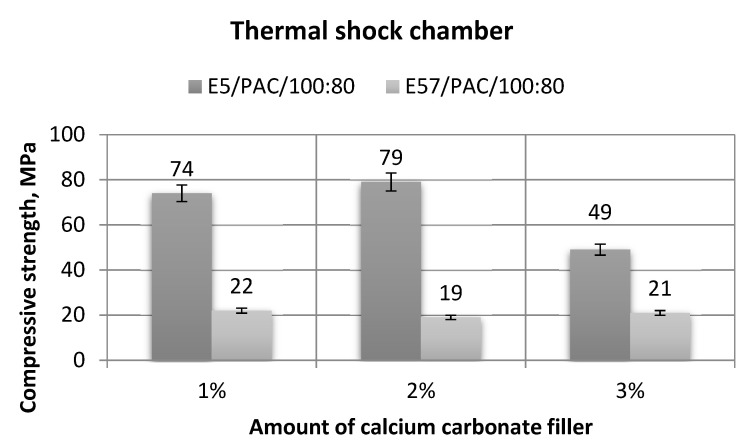
Comparison of the compressive strength between the E5/PAC/100:80 and E57/PAC/100:80 epoxy specimens modified by the calcium carbonate filler post-thermal shock chamber aging.

**Figure 21 materials-13-05439-f021:**
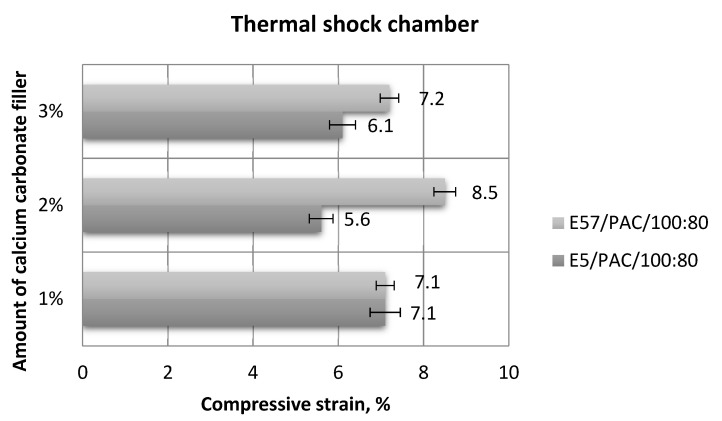
Comparison of the compressive strain between the E5/PAC/100:80 and E57/PAC/100:80 epoxy specimens modified by the calcium carbonate filler post-thermal shock chamber aging.

**Figure 22 materials-13-05439-f022:**
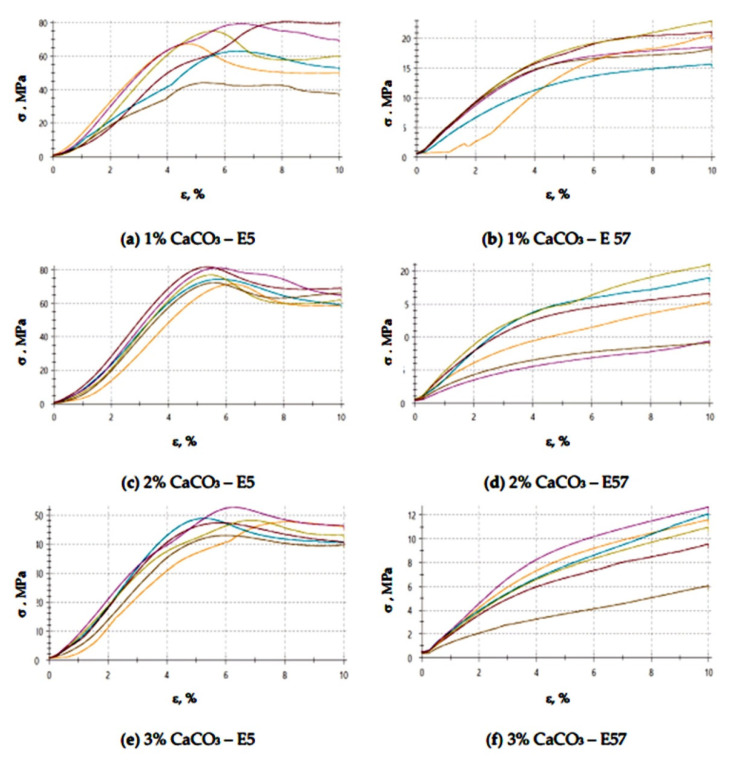
Stress–strain curves of the E5/PAC/100:80 and E57/PAC/100:80 epoxy specimens modified by the calcium carbonate filler post-thermal shock chamber aging: (**a**) 1% CaCO_3_—E5, (**b**) 1% CaCO_3_—E57, (**c**) 2% CaCO_3_—E5, (**d**) 2% CaCO_3_—E57, (**e**) 3% CaCO_3_—E5, (**f**) 3% CaCO_3_—E57.

**Figure 23 materials-13-05439-f023:**
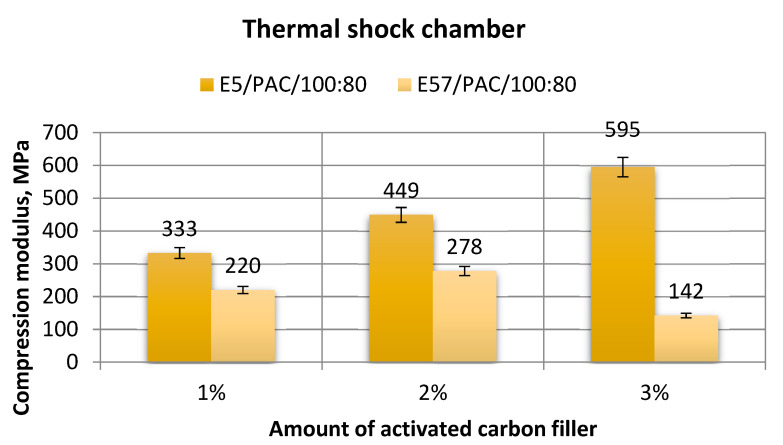
Comparison of the compression modulus between the E5/PAC/100:80 and E57/PAC/100:80 epoxy specimens modified by the activated carbon filler post-thermal shock chamber aging.

**Figure 24 materials-13-05439-f024:**
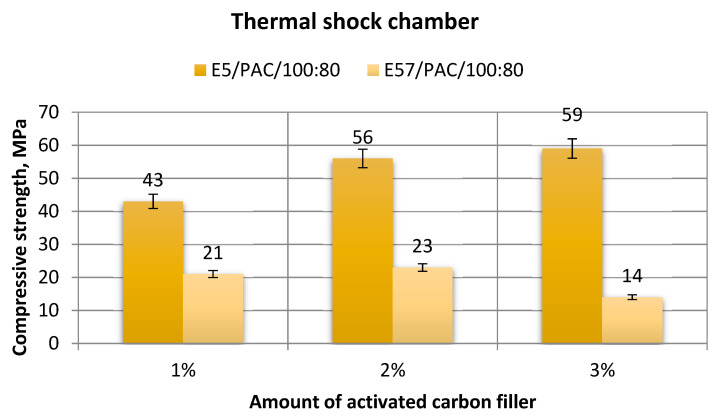
Comparison of the compressive strength between the E5/PAC/100:80 and E57/PAC/100:80 epoxy specimens modified by the activated carbon filler post-thermal shock chamber aging.

**Figure 25 materials-13-05439-f025:**
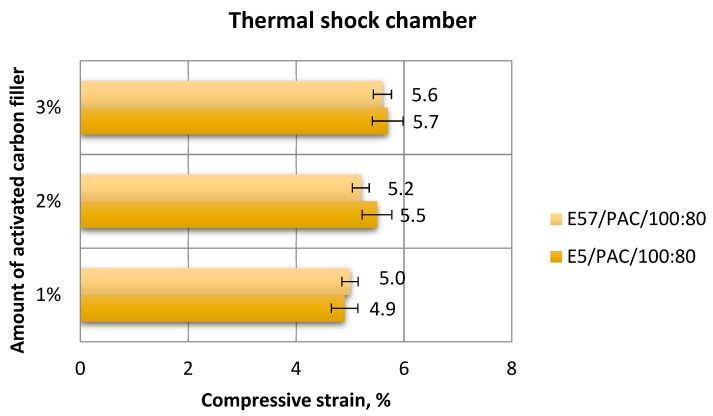
Comparison of the compressive strain between the E5/PAC/100:80 and E57/PAC/100:80 epoxy specimens modified by the activated carbon filler post-thermal shock chamber aging.

**Figure 26 materials-13-05439-f026:**
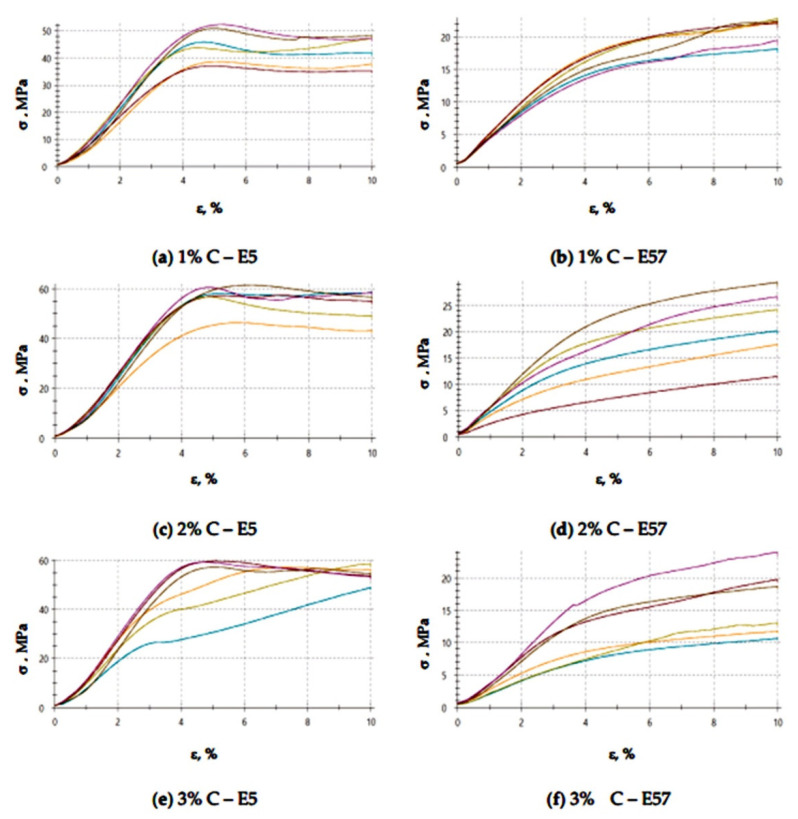
Stress–strain curves of the E5/PAC/100:80 and E57/PAC/100:80 epoxy specimens modified by the activated carbonate filler post-thermal shock chamber aging: (**a**) 1% C—E5, (**b**) 1% C—E57, (**c**) 2% C—E5, (**d**) 2% C—E57, (**e**) 3% C—E5, (**f**) 3% C—E57.

**Figure 27 materials-13-05439-f027:**
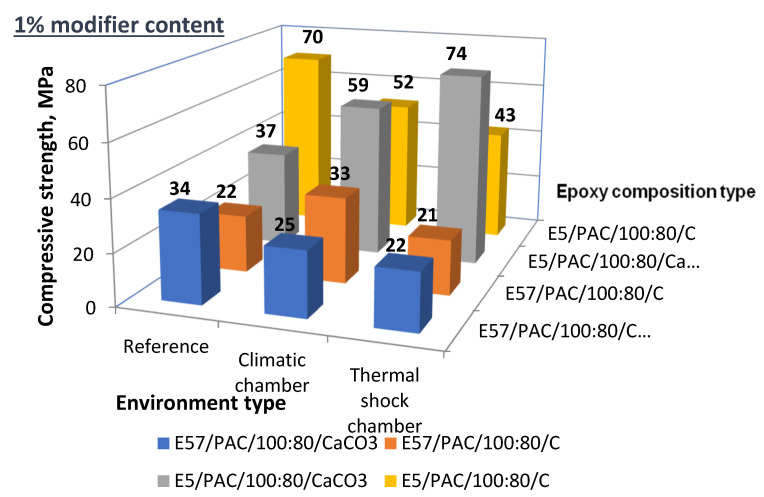
Compressive strength of the 1% filler epoxy compounds post-aging under different conditions.

**Figure 28 materials-13-05439-f028:**
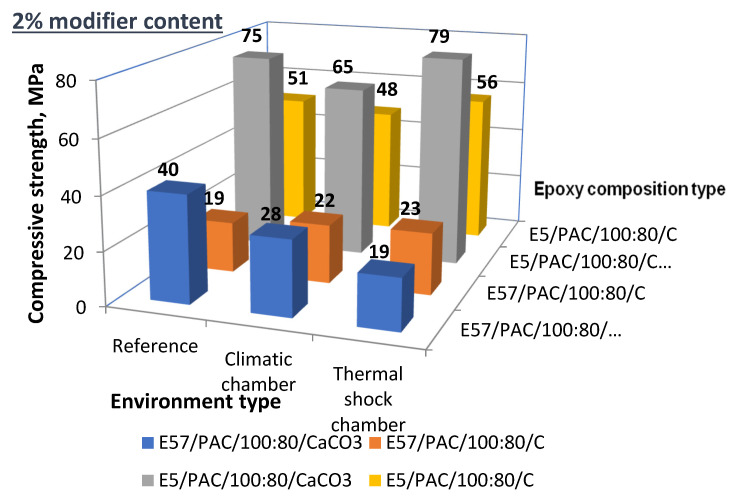
Compressive strength of the 2% filler epoxy compounds post-aging under different conditions.

**Figure 29 materials-13-05439-f029:**
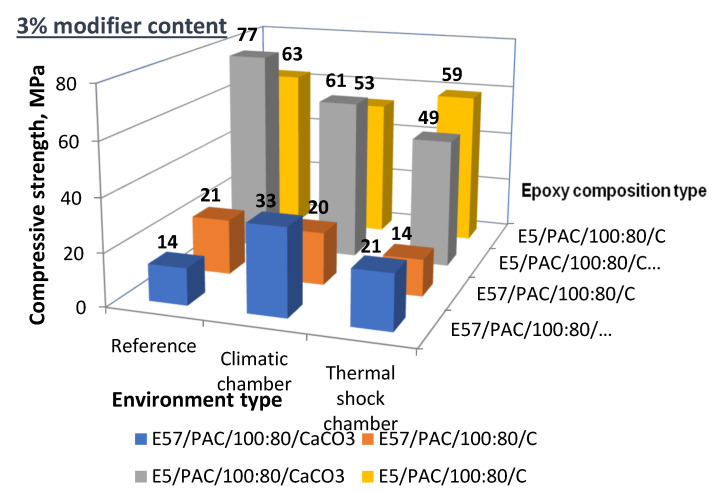
Compressive strength of the 3% filler epoxy compounds post-aging under different conditions.

**Table 1 materials-13-05439-t001:** Modified epoxy compounds with calcium carbonate (CaCO_3_) as the filler.

Resin (Trade Name)	Epoxy Resin/Curing Agent Weight Ratio	Curing Agent Type	Filler Percentage Ratio in 100 Parts by Weight of Epoxy Resin	Epoxy Compounds Designation
Epidian 5 (unmodified epoxy resin)	100:80	polyamide	1%	E5/PAC/100:80/CaCO_3_/1%
			2%	E5/PAC/100:80/ CaCO_3_/2%
			3%	E5/PAC/100:80/ CaCO_3_/3%
Epidian 57 (modified epoxy resin)	100:80	polyamide	1%	E57/PAC/100:80/ CaCO_3_/1%
			2%	E57/PAC/100:80/ CaCO_3_/2%
			3%	E57/PAC/100:80/ CaCO_3_/3%

**Table 2 materials-13-05439-t002:** Modified epoxy compounds with activated carbon (C) as the filler.

Resin (Trade Name)	Epoxy Resin/Curing Agent Weight Ratio	Curing Agent Type	Filler Percentage Ratio in 100 Parts by Weight of Epoxy Resin	Epoxy Compounds Designation
Epidian 5 (unmodified epoxy resin)	100:80	polyamide	1%	E5/PAC/100:80/C/1%
			2%	E5/PAC/100:80/C/2%
			3%	E5/PAC/100:80/C/3%
Epidian 57 (modified epoxy resin)	100:80	polyamide	1%	E57/PAC/100:80/C/1%
			2%	E57/PAC/100:80/C/2%
			3%	E57/PAC/100:80/C/3%

**Table 3 materials-13-05439-t003:** Number of reference specimens.

Reference Samples, Amount				
Amount of Filler	Epoxy Compounds			
	E5/PAC/100:80/CaCO_3_	E5/PAC/100:80/C	E57/PAC/100:80/CaCO_3_	E5/PAC/100:80/C
1%	6	6	6	6
2%	6	6	6	6
3%	6	6	6	6
Sum	18	18	18	18

**Table 4 materials-13-05439-t004:** Aging conditions of the epoxy specimens in the climate chamber.

Parameters	Type of Epoxy Compounds	
	E5/PAC/100:80/C E5/PAC/100:80/CaCO_3_	E57/PAC/100:80/C E57/PAC/100:80/CaCO_3_
Curing time	7 days	
Aging time	10 weeks	
Temperature	82 °C	
Humidity	95%	

**Table 5 materials-13-05439-t005:** Aging conditions of the epoxy specimens in the thermal shock chamber.

Parameters	Type of Epoxy Compounds	
	E5/PAC/100:80/C E5/PAC/100:80/CaCO_3_	E57/PAC/100:80/C E57/PAC/100:80/CaCO_3_
Curing time	7 days	
Aging time	10 weeks	
Temperature	+82 °C/−40 °C	
Time of thermal shock cycle	15 min	

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
