# Peer review of "Experimental Study of Mechanical Properties of Epoxy Compounds Modified with Calcium Carbonate and Carbon after Hygrothermal Exposure"

_materials, 2020, doi:10.3390/ma13235439_

Round 1

Reviewer 1 Report

For the composites of polymer and inorganic materials, the interfacial adhesion of polymer chains to the surfaces of inorganic materials plays a key role in the achievement of good mechanical properties. In order to achieve good interfacial adhesion, surface modification of inorganic materials is necessary. In this manuscript, the authors did not study the surfaces of CaCO3 and carbon. It seems they even did not realize the importance of the interfacial adhesion. They just simply measured the mechanical properties of the materials. They have a long way to go before they reach the right way. 

Author Response

Dear Sir or Madam,

I would like to thank the reviewer and the editor very much for their useful comments, which have helped me to improve the manuscript.

I hope that I have properly addressed their comments. All corrections are marked in color in the revised manuscript.

Reviewer #1

For the composites of polymer and inorganic materials, the interfacial adhesion of polymer chains to the surfaces of inorganic materials plays a key role in the achievement of good mechanical properties. In order to achieve good interfacial adhesion, surface modification of inorganic materials is necessary. In this manuscript, the authors did not study the surfaces of CaCO3 and carbon. It seems they even did not realize the importance of the interfacial adhesion. They just simply measured the mechanical properties of the materials. They have a long way to go before they reach the right way. 

Response:Thank you for your comment. The comments are very valuable and will be used as much as possible in future research. Indeed, the surface modification of inorganic materials In order to achieve good interfacial adhesion was not discussed, although the author encountered publications in which this aspect is discussed. Numerous methods have been studied for fiber surface treatment, such as electrochemical oxidation, oxidation in strong acids, plasma oxidation, and ozone oxidation. For example He at al. [Mixed resin and carbon fibres surface treatment for preparation of carbon fibres composites with good interfacial bonding strength, Materials and Design 31 (2010) 4631–4637] used anodic oxidation treatment to modify the surface of carbon fibres. They presented that surface analysis indicates that the amount of carbon fibre chemisorbed oxygen-containing groups, active carbon atom, the surface roughness, and wetting ability increases after treatment. Soo-Jin et al. [Electrochemical treatment on activated carbon fibers for increasing the amount and rate of Cr(VI) adsorption, Carbon 37 (1999) 1223–1226] used  anodic oxidation of carbon fibers to improve the adhesive properties of carbon fibers whereby negative ions were attracted to the surface of the a resin matrix to increase the interlaminar shear strength.Some authors focused on the effect of different nano-fillers on the mechanical properties and the joint strength of epoxy are reported, with a review a techniques for nano-fillers dispersion (e.g. Jojibabu et al., Int J Adhes Adhes 96 (2020) 102454). I believe that the issue of filler surface preparation indicated by the Reviewer is extremely important and requires further work. The article does not focus on this issue. In addition, the interfacial adhesion aspect is also related to the mixing process - the type and technological parameters of this process. This aspect was discussed, among others in the works of F.-L. Jin and S.-J. Park, also P. Jojibabu et al. I would like to mention that an article is being prepared, which presents various methods of mixing the components that are components of the analyzed epoxy compositions.The focus was on presenting the results of strength tests, which is important from the point of view of engineering and indeed, an important aspect indicated by the reviewer was omitted. The higher for example the compressive strength value, the greater the load on the material can be, therefore, the focus was on determining selected mechanical properties.Some works and some information have been included in the text. I tried to make corrections to both the method and the results, which was noted in the review form.

Yours faithfully,

Anna Rudawska

Reviewer 2 Report

In this study, the authors have reported a composite consisting of calcium carbonate or carbon fillers within an epoxy matrix and found an increase in mechanical performance in these composite systems. Currently, I cannot recommend the acceptance of the paper in this form since statements scattered throughout the manuscript show that the authors lack the fundamental understanding of how composites form, and also how engineered composite functions. Included below are some major comments which might help the authors improved their manuscript:

Major suggestions:

  1. The use of calcium carbonate (or carbon) dispersed in epoxy matrix is not really new. The introduction section does not develop enough work done in the field. I think about Kaixi Li's work who presented nano-sized calcium carbonate as a reinforcing agent to have good mechanical performance of epoxy matrices. Those work should presented.
  2. Moreover, crosslinking reaction of calcium carbonate/epoxy resin was also reported by the past by Soo-Jin Park et al. That work should be cited too. I suggest the author to complete the referencing .... and to reinforce the position of his work compared to the state of the art.
  3. The authors should include the data and explanations for all the weight ratios or reason for explaining physical properties with specific ratio should be included.
  4. For the tensile strength test, what is the size of the sample? More details are needed for better understanding of this magnitude?
  5. Is it possible to show the fractured cross-session of the composites? The morphology of the fractured cross-secession may provide more information about the fracture type, brittle or ductile.
  6. Raman spectroscopy and TEM analysis should be exploited for assessing the surface structure and morphology of the obtained nanocomposites.
  7. The resulting stress-strain curves should be provided in the manuscript. In addition, authors do not compare their results of mechanical measurements with preexisting literature results.

Author Response

Dear Sir/Madam

I would like to thank the reviewer and the editor very much for their useful comments, which have helped me to improve the manuscript.

I hope that I have properly addressed their comments. All corrections are marked in color in the revised manuscript.

Reviewer 2

In this study, the authors have reported a composite consisting of calcium carbonate or carbon fillers within an epoxy matrix and found an increase in mechanical performance in these composite systems. Currently, I cannot recommend the acceptance of the paper in this form since statements scattered throughout the manuscript show that the authors lack the fundamental understanding of how composites form, and also how engineered composite functions.

Response:Thank you for your valuable comments.

Included below are some major comments which might help the authors improved their manuscript:

Major suggestions:

1. The use of calcium carbonate (or carbon) dispersed in epoxy matrix is not really new. The introduction section does not develop enough work done in the field. I think about Kaixi Li's work who presented nano-sized calcium carbonate as a reinforcing agent to have good mechanical performance of epoxy matrices. Those work should presented.

 Response:

Thank you for your valuable comments. The Kaixi Li's works and some information have been included in the text. I would like to add that the interesting work of Hongwei He and Kaixi Li and other authors has already been cited in the manuscript as reference [12].

2. Moreover, crosslinking reaction of calcium carbonate/epoxy resin was also reported by the past by Soo-Jin Park et al. That work should be cited too. I suggest the author to complete the referencing .... and to reinforce the position of his work compared to the state of the art.

 Response:

Thank you for your valuable comments. The Soo-Jin Park et al. articles and some information have been included in the text. The referencing has been completed.

3. The authors should include the data and explanations for all the weight ratios or reason for explaining physical properties with specific ratio should be included.

Response:Thank you for your comment. The information has been included to the text. The epoxy compounds o was set at 100:80 resin:polyamide curing agent (in weight), corresponding to the stoichiometric epoxy/amide molar ratio. Based on the information presented in the works [11,13,21-23, 33,35], it was noticed that scientists use a different amount of the calcium carbonate filler. According to the literature data, the amount of filler may be from 2 to 8% parts by weight. He at al. [23] underlined that performance of polymeric materials can be improved by introducing some small amount of filler < 5 wt.%. Yang et al. [33] pointed out that the mechanical properties of the epoxy composites were improved by increasing the amount of CaCO3 added but decreased when the filler content reached 2%. Besides Maisel and Waso [35] mentioned that percentage amount of silica with carbon carbonate fillers was limited to a maximum of 3%. The addition of 1 g, 2 g and 3 g of fillers per 100 g of epoxy resin was used in the experiments to prepare the epoxy compounds. The same amount of addition of both fillers was assumed for comparison purposes.

4. For the tensile strength test, what is the size of the sample? More details are needed for better understanding of this magnitude?

Response: Thank you for your comment. The size of the sample was presented in the 2.3 point. The epoxy compound specimen dimensions after curing process were: diameter, d = 15±0.5 mm and length, L = 40±1 mm. It was also presented in Fig. 1. To determine the modulus of elasticity in compression, a cylindrical sample can be used, and the ratio of the height of the sample (h) to its diameter (d) should be 2 to 4, it is recommended that a diameter of 15 mm be used for polymeric materials. The specified conditions were performed during the preparation of samples for compressive strength tests. 

5. Is it possible to show the fractured cross-session of the composites? The morphology of the fractured cross-secession may provide more information about the fracture type, brittle or ductile.

 Response:Thank you for your comment. At the moment, I cannot take photos of the damaged samples, due to the fact that I provide remote work (as well as other employees) due to the orders of the authorities in connection with the epidemic situation in my country. Due to these recommendations, I have limited access to the department and I apologize for such a explanation, but I want to present the situation fairly. But the general sentences describing the nature of the destruction of the samples were presented (The visual inspection of the behaviour of the specimens during the strength tests demonstrated that unmodified epoxy resin (Epidian 5) combined with PAC produced an epoxy compounds which frequently cracked during leveling of the edges and in the strength tests. The epoxy specimens made from E57/PAC/100:80 were much more elastic. It was necessary to fabricate several lots of the specimens, as their edges suffered from strong deformations when leveled. The ageing of the specimens improved their hardness). Although no detailed analysis of the failure of the epoxy composition samples after the shear strength tests has been provided, it can be generally stated that it was observed that the failure of the samples in most cases showed brittle failure characteristics. It was also noticed in other works by the author.Indeed the morphology of the fractured cross-secession may provide more information about the fracture type.I agree with this valuable remark.

6. Raman spectroscopy and TEM analysis should be exploited for assessing the surface structure and morphology of the obtained nanocomposites.

 Response:Thank you for your helpful comment. Obviously, Raman spectroscopy and TEM analysis would be helpful for assessing the surface structure and morphology of the obtained nanocomposites. Unfortunately, the work did not focus on this aspect. However, in the part describing the prepared samples, I have included exemplary SEM images, which partially show the structure of the epoxy compositions, including the degree of filler dispersion in the epoxy matrix.

7. The resulting stress-strain curves should be provided in the manuscript. In addition, authors do not compare their results of mechanical measurements with preexisting literature results.

Response:Thank you for your comment. The examples stress-strain curves were provided in the manuscript. Comparative results of sample elongation are presented in the existing graphs. I hope the reviewer will accept this form of presentation.  I would like to add that after the comparison results I try to compare obtained results with results other researches. Perhaps the reviewer considered this comparison insufficient, so this part of the article has been supplemented, as suggested by the reviewer.         

Yours faithfully,

Anna Rudawska            

Reviewer 3 Report

This study deals with the effect of hygrothermal exposure on the compressive properties of samples made of epoxy resins modified with calcium carbonate or carbon fillers. Two types of hygrothermal conditions were tested: ageing at high temperature and high relative humidity in a climate chamber for 10 weeks and exposure to thermal shocks with temperatures varying between -40°C and 82°C. The compression properties of the epoxy specimens were measured before and after hygrothermal exposures. This study is interesting for several fields of applications (packaging, composites, adhesives, coatings) where epoxy resins are used.

In general, it is difficult to carry out hygrothermal exposure tests on materials because it requires long-duration experiments. This was done in this study. Interesting results have been gained. However, the results are not discussed enough. The link between the filler types and the epoxy resin types and the compound properties is not explained. In addition, the effect of ageing as a function of the filler types and the epoxy types and the hygrothermal conditions is not explained. Furthermore, some complementary analyses should be performed to better discuss the observed trends.

For instance, specimens made of epoxy only should be tested in the same conditions as the samples made of modified epoxy resins. This would help to understand the effects of hygrothermal exposures obtained using modified epoxy resins.

The microstructure of specimens made of epoxy resins modified with calcium carbonate and carbon fillers is not studied. Thus it is difficult to understand the evolution of the mechanical properties with the filler amount. Are there filler aggregates in the specimens containing the highest amount of fillers? The microstructures of specimens should be studied before and after hygrothermal exposures.

The thermophysical properties of both tested epoxy resins should be investigated, using for instance DSC. What is the degree of during reaction with the used manufacturing reaction and the hygrothermal exposures?

The author should present the compound specific properties. This would allow a better comparison of the mechanical performances of samples, considering their density.

Other remarks:

Line 2 », abstract : no need to cite here the standard used.

Line 69 : co tent à content

Line 70 : such à such as

Line 71 : such à such as  and is drecreased à are decreased

Introduction : what are the typical dimensions of calcium carbonate fillers and nano fillers ? Same question for the carbon fillers ? And also, what is the typical shape of these particles ?

Line 82 : what this limit of 3% in the study of Maisel and Waso ?

Line 85 : what is the physical or chemical origin of this enhancement in the ageing properties due to carbon fillers ?

Line 100 : environment à environments

Line 108 : adhesives à adhesive

Line 124 : Compoundted à compounded

Line 202 : please describe this specific strength fixture.

Line 223 : this sentence is unclear. Please rewrite.

Line 228 : why such a difference between two types of fillers ? Aggregates of calcium carbonate fillers ?

Table 5 : what is the “kinetics” of thermal shocks ?

Line 435 : The epoxy version did not… : this sentence is not clear. What is the epoxy version ?

Author Response

Dear Sir/Madam,

I would like to thank the reviewer and the editor very much for their useful comments, which have helped me to improve the manuscript.

I hope that I have properly addressed their comments. All corrections are marked in color in the revised manuscript.

I would like to inform you that I am sending the answer in the attachment.

Yours faithfully

Round 2

Reviewer 1 Report

The quality of this manuscript has improved significantly, and this manuscript can be accepted for publication now. 

Author Response

Reviewer #1

Dear Reviewer,

I would like to thank you very much for your very valuable comments, which have helped me to improve the manuscript.

Best regards,

Anna Rudawska

Reviewer 2 Report

In reviewing the revised manuscript and point-by-point response to reviewers' comments, the authors have addressed my concerns and thus this revised manuscript would be suitable for publication in materials.

Author Response

Reviewer #2

Dear Reviewer,

I would like to thank you very much for your very valuable comments, which have helped me to improve the manuscript.

Best regards,

Anna Rudawska

Reviewer 3 Report

The author has made significant changes to the manuscript.

The new compression stress-strain curves are interesting. However, for some samples, these curves exhibit a large discrepancy and it is difficult to understand how the author could extract from  the various mechanical parameters that are shown in the study and associate to these data such small error bars.

The discrepancy between these curves should be commented. Given that the authors performed compression tests, one could wonder if some samples did exhibit buckling phenomena or other uncontrolled kinematical conditions that could be at the origin of such a discrepancy.

Author Response

Reviewer #3

Dear Reviewer,

I would like to thank you very much for your very valuable comments, which have helped me to improve the manuscript.

Best regards,

Anna Rudawska
